# The instrument constant of sky radiometer (POM-02), Part I: Calibration constant

Akihiro Uchiyama[1], Tsuneo Matsunaga[1], Akihiro Yamazaki[2]

[1] Center for Global Environmental Research, National Institute for Environmental Studies, Tsukuba, Ibaraki, 305-8506, Japan

[2] Meteorological Research Institute, Japan Meteorological Agency, Tsukuba, Ibaraki, 305-0052, Japan

*Correspondence to*: Uchiyama Akihiro (uchiyama.akihiro@nies.go.jp)

## Abstract

Ground-based networks have been developed to determine the spatiotemporal distribution of the optical properties of aerosols using radiometers. In this study, the precision of the calibration constant ($V_0$) for the sky radiometer (POM-02) which is used by SKYNET was investigated. The temperature dependence of the sensor output was also investigated, and the dependence in the 340, 380, and 2200 nm channels was found to be larger than for other channels, and varied with the instrument. In the summer, the sensor output had to be corrected by a factor of 1.5 to 2% in the 340 and 380 nm channels and by 4% in the 2200 nm channel in the measurements at Tsukuba ((36.05°N, 140.13°E), with a monthly mean temperature range of 2.7 to 25.5 ºC). In the other channels, the correction factors were less than 0.5%. The coefficient of variation (CV, standard deviation/mean) of $V_0$ from the normal Langley method based on the data measured at the NOAA Mauna Loa Observatory is between 0.2 and 1.3%, except in the 940 nm channel. The effect of gas absorption was less than 1% in the 1225, 1627, and 2200 nm channels. The degradation of $V_0$ for wavelengths shorter than 400 nm (−10 to −4% per year) was larger than that for wavelengths longer than 500 nm (−1 to nearly 0% per year). The CV of $V_0$ transferred from the reference POM-02 was 0.1 to 0.5%. Here, the data were simultaneously taken at 1-minute intervals on a fine day,

and data when the airmass was less than 2.5 were compared. The $V_0$ determined by
the improved Langley (IML) method had a seasonal variation of 1 to 3%. The RMS
error from the IML method was about 0.6 to 2.5%, and in some cases, the maximum
difference reached 5%. The trend in $V_0$ after removing the seasonal variation was
almost the same as for the normal Langley method. Furthermore, the calibration
constants determined by the IML method had much higher noise than those
transferred from the reference. The modified Langley method was used to calibrate the
940 nm channel with onsite measurement data. The $V_0$ obtained with the modified
Langley method compared to the Langley method was 1% more accurate on stable and
fine days. The general method was also used to calibrate the shortwave-infrared
channels (1225, 1627, and 2200 nm) with onsite measurement data. The $V_0$ obtained
with the general method differed from that obtained with the Langley method of $V_0$
by 0.8, 0.4, and 0.1% in December 2015, respectively.
**1. Introduction**
Atmospheric aerosols are an important constituent of the atmosphere. Aerosols
change the radiation budget directly by absorbing and scattering solar radiation and
indirectly through their role as cloud condensation nuclei (CCNs), thereby increasing
cloud reflectivity and lifetime (e.g., Ramanathan et al. 2001, Lohmann and Feichter
2005). Aerosols also affect human health as one of the main components of air pollution
(Dockery et al. 1993, WHO 2006, 2013).
Atmospheric aerosols have a large variability in time and space. Therefore,
measurement networks covering an extensive area on the ground and from space have
been developed and established to determine the spatiotemporal distribution of
aerosols.
Ground-based observation systems, such as those using radiometers, are more
reliable and easier to install and maintain than space-based systems. Therefore,
ground-based observation data are used to validate data obtained from space-based
systems (Kahn et al. 2005, Remer et al. 2005, Mélin et al. 2010). Well-known
ground-based networks include AERONT (AErosol RObotic NETwork) (Holben et al.
1998), SKYNET (Takamura and Nakajima 2004), and PFR-GAW (Precision Filter
Radiometer-Global Atmosphere Watch) (Wehrli 2005).
In ground-based observation networks, direct solar irradiance and sky radiance are
measured, and the column average effective aerosol characteristics are retrieved by
analyzing these data: optical depth, single scattering albedo, phase function, complex
refractive index, and size distribution. To improve the measurement accuracy, it is
important to know the characteristics of the instruments and to calibrate the
instruments. Furthermore, from the view point of the validation of optical properties
retrieved from the satellite measurement data, it is important to know the magnitude
of the error in the ground-based measurements.
In SKYNET, the radiometers POM-01 and POM-02, manufactured by Prede Co. Ltd.,
Japan, are used. These radiometers are called 'sky radiometers', and measure both the
solar direct irradiance and sky-radiances (Takamura et al. 2004). The objectives in this
study are to investigate the current status of and problems with the sky radiometer.
There are two constants that we must determine to make accurate measurements.
One is the calibration constant, and the other is the solid view angle (SVA) of the
radiometer. Following Nakajima et al. (1996), this paper uses the SVA to quantify the
magnitude of the field of view (FOV). The calibration constant $V_0$ is the output of the
radiometer to the extra-terrestrial solar irradiance at the mean earth-sun distance (1
astronomical unit (AU)) at the reference temperature. The SVA is a constant which
relates the sensor output to the sky radiance. The ambient temperature affects the
sensor output, and this temperature dependence must be considered when analyzing
data from POM-01 and POM-02 (Prede, Japan). In this study, the temperature
dependence of POM-02 and the calibration of the sensor are described. The SVA is
described in detail in Part II.
In section 2, we briefly describe the data used in this study. In section 3, firstly, the
temperature characteristics of POM-02 are described. Though the majority of POM-01
and POM-02 users do not explicitly consider the temperature dependence of the
instruments, some channels have a large temperature dependence.
Secondly, the precision of the calibration constant is described. Most POM-01 and
POM-02 users calibrate the sky radiometers with the Improved Langley (IML) method
(Tanaka et al. 1986, Campanelli et al. 2004), because this method only needs on-site
measurement data and special measurements for calibration are not required. One of
the goals of this paper is to examine the difference between the $V_0$ obtained by the
IML method and by the normal Langley method, but before that, in section 4, we
briefly review the Langley method, and consider the precision of the normal Langley
method using the data obtained at the NOAA Mauna Loa Observatory (MLO), which is
one of the most suitable places for sky radiometer calibration by the normal Langley
method, and the precision of the calibration constant transfer obtained from
side-by-side measurement. In section 5, we briefly review the IML method, and though
Campanelli et al. (2004) have already estimated the RMS error of the IML method, we
estimate it again and show the time variation and the relation between the calibration
constant and temperature dependence. Then, in section 6, an example of the precision
of the calibration using a calibrated integrating sphere is shown.
In SKYNET, the 940, 1627, and 2200 nm channels were not used. Therefore, the
precipitable water vapor (PWV) and the optical depth at 1627 nm are not estimated.
However, these parameters are estimated in AERONET. In sections 7 and 8,
calibration methods for these channels are shown using on-site measurement data. In
section 9, the results are summarized.

**2. Data**
In this study, measurements were conducted using two POM-02 sky radiometers
which are used by the Japan Meteorological Agency/Meteorological Research Institute
(JMA/MRI). One is used as a calibration reference, POM-02 (Calibration reference),
and the other is used for continuous measurement at the Tsukuba MRI observation
site, POM-02 (Tsukuba). In Table 1, the nominal specifications of the filters are shown.
The JMA/MRI does not use the 315 nm channel because the transmittance of the lens
was low at this wavelength. Instead, the JMA/MRI added a 1225 nm channel. The
sensor output in the file storing the measurement values of POM-02 is the current: the
unit is Ampere (A). Therefore, the unit of the calibration constant in this paper is
Ampere (A).
To calibrate the reference POM-02 by the normal Langley method (i.e., the same
airmass of air molecule scattering for all attenuating substances, see section 4.1), the
measurements were conducted at the NOAA Mauna Loa Observatory (MLO) for about
one month every year, for more than twenty years. The MLO (19.5362°N, 155.5763°W)
is located at an elevation of 3397.0 meters amsl on the northern slope of Mauna Loa,
Island of Hawaii, Hawaii, USA. The atmospheric pressure is about 680 hPa. The MLO
is one of the most suitable places to obtain data for a Langley plot (Shaw 1983) and for
a solar disk scan. Using these data, the calibration constant is estimated and the SVA
is calculated.
The continuous observation was performed at the JMA/MRI (36.05°N, 140.13°E) in
Tsukuba, which is located about 50 km northeast of Tokyo. Using these continuous
measurement data, the calibration constants for the IML method were calculated
using the SKYRAD software package (Nakajima et al. 1996, OpenCLASTR,
http://www.ccsr.u-tokyo.ac.jp/~clastr/). Usually, the calibration of POM-02 for
continuous measurement is conducted by comparison with the side-by-side
measurement data from the reference POM-02.
The temperature dependence of the sensor output was measured using the same
equipment which was originally used to measure the temperature dependence of the
pyranometer. This equipment is managed and maintained by a branch of the JMA
Observation Department. The main components of this equipment are a
temperature-controlled chamber, light source, and stabilized power supply.
The measurements for investigating the temperature characteristics of POM-02
were made as follows.
To stabilize the equipment, the power supply of the equipment was turned on the
day before the measurement date. On the measurement day, first the light source was
turned on, then the temperature was varied every 90 minutes, and the temperature
and output from POM-02 were recorded continuously. The temperature was set to 40,
20, 0, −20, 0, 20, 40, and 20 ºC. It took about 30 (40) minutes after increasing
(decreasing) the temperature for the temperature and the output of POM-02 to become
stable. Temperature characteristics were investigated using data between 70 and 90
minutes after varying the temperature.
To check the stability of the equipment, the staff of the JMA recorded the output of
the pyranometer CMP-22 (Kipp & Zonen, Netherland) continuously for 11 hours at a
temperature setting of 20 ºC. As a result, the variation of the hourly mean values of the
output was within ±0.05%.
The temperature correction was performed for each individual measurement value.
The temperature dependence of the sensor output was approximated by the following
equation:
$$V(T)/V(T = Tr) = 1.0 + C_1(T - Tr) + C_2(T - Tr)^2 \qquad\qquad (1)$$
where $V(T)$ is the sensor output at temperature $T$, $V(T = Tr)$ is the sensor output
at reference temperature $Tr$, and coefficients $C_1$ and $C_2$ were determined by the
least squares method. In the case of POM-02, the sensor output is current, and the unit
is Ampere (A). Therefore, the measured $V(T)$ is corrected using eq. (1).

**3. Temperature dependence of sensor output**
In this section, the temperature characteristics of the POM-02 are described. The
POM-02 is temperature-controlled. However, the temperature control is insufficient.
Therefore, the sensor output of the POM-02 is dependent on the environmental
temperature.
The purpose of the temperature control is to keep the temperature inside the
instrument from decreasing below levels which will reduce the instrument's precision.
Instruments are designed to activate the heater when the inside temperature is less
than 20 or 30 ℃. For colder regions such as polar regions, the minimum temperature
threshold for activating the heater is 20 ℃, and in other regions, the threshold is 30 ℃.
When the temperature near the rotating filter wheel inside the instrument is below
the threshold temperature (20 or 30 ℃), the instrument is heated. When the
temperature exceeds the threshold, heating is stopped. However, there is no cooling
mechanism for when the temperature inside the instrument is higher than its
threshold temperature. To monitor the temperature inside the instrument, a
temperature sensor is attached near the rotating filter wheel. Furthermore, the
shortwave-infrared detector, which is thermoelectrically cooled, is equipped with a
temperature sensor and temperature data can be recorded.

In Fig. 1, an example of the relation between the temperature near the rotating filter

wheel and the environmental temperature for POM-02 (Calibration Reference) is
shown. The red line is the temperature near the rotating filter wheel that holds the
individual filters and the blue line is the temperature of the shortwave-infrared
detector. The temperature control setting of this POM-02 is 20 ℃. Since heat is
generated from the electric circuit inside the POM-02, the inside temperature exceeds
20 ℃ even if the ambient temperature is less than 20 ℃. The heater stops when the
inside temperature of the POM-02 exceeds 20 ℃. However, since there is no cooling
mechanism, the temperature inside the POM-02 rises as the ambient temperature
increases. When the ambient temperature is very low, the temperature does not rise to
20 ℃ because the heater is not powerful enough. For example, when the ambient
temperature was about −20 ℃, the internal temperature was about 0 ℃. The ambient
temperature was varied in the order of 40, 20, 0, −20, 0, 20, 40, 20 ℃. Since the
mounting position of the temperature sensor and the thermal structure of the
instrument were different for each product, not every POM-02 temperature responds
in the same way.

In Fig. 2, the relation between the sensor output and the inside temperature near

the filter wheel for POM-02 (Calibration Reference) is shown. The sensor output is
normalized by the sensor output at 20 ℃. The ambient environmental temperature
was varied from −20 to 40 ℃. The detector used for wavelengths shorter than 1020 nm
was a Si photodiode, and the detector for the 1225, 1627, and 2200 nm wavelengths
was a thermoelectrically cooled InGaAs photodiode. In this study, the former
wavelength region is referred to as the "visible and near-infrared region" and the latter
one is the "shortwave-infrared region".

The temperature dependence of the sensor output in the 340 and 2200 nm channels

was larger than in the other channels. The range of the atmospheric temperature at
Tsukuba was about −5 to 35 ºC (the range of the monthly mean temperature was 2.7 to
25.5 ºC), and the resulting inside temperatures were between 15 and 35 ºC (Fig. 1), and
the change in the instrument response was less than 1.5% except for the 340 and 2200
nm channels. The temperature dependence of the sensor output varies with the
channel.

In the 340 nm channel, the sensor output decreased by 7% when the internal

temperature increased from 20 to 40 ºC. In the 2200 nm channel, the sensor output
decreased by a rate of 5 to 6% per 10 degrees of temperature increase. Therefore, the
temperature dependence of the sensor output cannot be ignored in these two channels.

In Fig. 3, the temperature dependence of the sensor output for POM-02 (Tsukuba) is

shown. The temperature dependence of the sensor output in the 340 nm (2200 nm)
channel for this POM-02 is larger (smaller) than that for the calibration reference
POM-02. In the 340 and 380 nm channels, the rate of sensor output decrease was
about 1.5% per 10 degrees, and in the 2200 nm channel, the rate of sensor output
decrease was about 3% per 10 degrees. In the other channels, the temperature
dependence of the sensor output was less than 1% for temperatures between 0 and 40
ºC.

The temperature dependence of the detector sensitivity as shown in the

specifications data sheet of the detector
(https://www.hamamatsu.com/resources/pdf/ssd/s1336_series_kspd1022e.pdf) is
almost zero (indistinguishable from zero in the sensitivity diagram) at wavelengths
from 300 nm to 950 nm. At a wavelength of 1020 nm, it is about 0.2% per degree. At
wavelengths of 1225 nm, 1627 nm, and 2200 nm, they are almost zero, −0.05% per
degree, and 0.02% per degree, respectively
(https://www.hamamatsu.com/resources/pdf/ssd/g12183_series_kird1119e.pdf). The
temperature dependencies of the sensor output shown in Figs. 2 and 3 are
characteristic of the entire instrument. Some channels exhibit greater temperature
dependence than the temperature dependence of the detector.

Though only two examples were shown here, the temperature dependence of the

sensor output differed between instruments. If we want to determine the temperature
dependence of the sensor output precisely, we need to measure it for each instrument
or only use channels with a small temperature dependence.

**4. Langley method**

In this section, the Langley method is briefly reviewed, and the Langley method

used in this study is described. Before investigating the RMS error of the IML method,
first the precision of the normal Langley method and the transfer of the calibration
constant are investigated. The transferred calibration constant can be obtained by
comparing side-by-side measurements of the direct solar irradiance.

**4.1 Brief review of Langley method**
According to the Beer-Lambert-Bouguer attenuation law, the directly transmitted
monochromatic solar irradiance $F(\lambda)$ at wavelength $\lambda$ is
$$F(\lambda) = \frac{F_0(\lambda)}{R^2}\exp(-\int_{z_0}^{\infty}k(\lambda,s)ds) \qquad (2)$$
where $F_0(\lambda)$ is the monochromatic solar irradiance at wavelength $\lambda$ at the mean
earth-sun distance (1 AU), $R$ is the earth-sun distance in AU, and $k(\lambda,s)$ is the
total spectral extinction coefficient at position $s$. The integral of $k(\lambda,s)$ is the optical
path length, and the integration is done along the path of the solar beam. In eq. (3)(2),
several atmospheric components contribute to $k(\lambda,s)$ : Rayleigh scattering by air
molecules, extinction by aerosol and cloud particles, absorbing gas such as water vapor
and ozone, and so on.
When the extinction coefficient is composed of several components, eq. (2) becomes
$$F(\lambda) = \frac{F_0(\lambda)}{R^2}\exp(-\sum_{i}\int_{z_0}^{\infty}k_i(\lambda,s)ds) \; . \qquad (3)$$
Introducing the vertical optical thickness (or optical depth) for each component,
$$\tau_i(\lambda) = \int_{z_0}^{\infty}k_i(\lambda,z)dz \qquad (4)$$
where the extinction coefficient for the $i$th component is integrated in the vertical
direction (Liou 2002).
Using the optical depth $\tau_i(\lambda)$, the optical path length is written as follows:
$$m_i(\theta)\tau_i(\lambda) = \int_{z_0}^{\infty}k_i(\lambda,s)ds \qquad (5)$$
where $m_i(\theta)$ is the airmass for the $i$th component, and $\theta$ is the solar zenith angle.
The airmass varies with the solar zenith angle and for small $\theta$ may be approximated
by $1/\cos(\theta)$. For large zenith angles ($\theta > 60°$), the sphericity and atmospheric
refraction must be taken into account. As $m_i(\theta)$ also depends on the vertical
distribution of a component, $m_i(\theta)$ is different for each component.
Substituting eq. (5) into eq. (3) gives the following:
$$F(\lambda) = \frac{F_0(\lambda)}{R^2}\exp(-\sum_i m_i(\theta)\tau_i(\lambda)).$$    (6)
Traditionally, the directly transmitted solar irradiance is represented as follows:
$$F(\lambda) = \frac{F_0(\lambda)}{R^2}\exp(-m_T(\theta)\tau_T(\lambda))$$    (7)
where $m_T(\theta)$ is the total airmass and $\tau_T(\lambda)$ is the total optical depth.
To obtain a measurable radiometer signal, $F(\lambda)$ is measured with some small but
non-zero finite bandwidth at the selected wavelength and finite field of view (Show,
1982). Spectral filter radiometers with a bandwidth of about 10 nm or less in the
visible and near infrared region were recommended and used for accurate
measurements (Shaw 1976, 1982, Reagan et al. 1986, Bruegge et al. 1992, Schmid and
Wehrli 1995, Holben et al 1998, Kazadzis et al. 2017).
The solar direct irradiance spectrally averaged by the spectral response function is
written as follows:
$$\overline{F}(\lambda_0) = \int_{\Delta\lambda} \phi(\lambda)\frac{F_0(\lambda)}{R^2}\exp(-\sum_i m_i(\theta)\tau_i(\lambda))d\lambda \Big/ \int_{\Delta\lambda} \phi(\lambda)d\lambda$$    (8)
where $\overline{F}(\lambda_0)$ is the solar direct irradiance spectrally averaged at the center
wavelength $\lambda_0$, and $\phi(\lambda)$ is the filter response function.
Since the wavelength dependence of the molecular scattering coefficient, extinction
coefficient by aerosols, and continuous absorption coefficient by gas are small, these
values are approximated by the value at the center wavelength $\lambda = \lambda_0$. The
extra-terrestrial solar irradiance is approximated by the filter-weighted value.
However, in the gas absorption band composed of many absorption lines, such as the
940 nm channel, the filter-weighted transmittance does not follow the
Beer-Lambert-Bouguer attenuation law,
$$\overline{F}(\lambda_0) = \frac{\overline{F}_0(\lambda_0)}{R^2}\exp(-\sum_i m_i(\theta)\tau_i(\lambda_0))\overline{T}_{gas}(\lambda_0,\theta)$$    (9)
where
$$\overline{T}_{gas}(\lambda_0,\theta) = \int_{\Delta\lambda} \phi(\lambda)\exp(-m_{gas}(\theta)\tau_{gas}(\lambda))d\lambda \Big/ \int_{\Delta\lambda} \phi(\lambda)d\lambda$$    (10)
and $\tau_{gas}$ is the optical depth of the gas absorption lines.
When estimating the optical depth of the aerosol from measurement of the direct solar
irradiance, the wavelength range where the absorption by gas is as small as possible is
chosen (Show 1982). When estimating the precipitable water vapor, a wavelength
range of 940 nm is often chosen.

Considering molecular scattering, absorption by ozone (Cappuis bands, Huggins

bands), extinction by aerosol, and absorption by gas absorption lines, eq. (9) becomes as
follows:

$$\bar{F}(\lambda_0) = \frac{\bar{F}_0(\lambda_0)}{R^2} \exp(-m_R(\theta)\tau_R(\lambda_0) - m_{O3}(\theta)\tau_{O3}(\lambda_0) - m_{aer}(\theta)\tau_{aer}(\lambda_0))\bar{T}_{gas}(\lambda_0,\theta)$$
$$= \frac{\bar{F}_0(\lambda_0)}{R^2} \exp(-m_T(\theta)\tau_T(\lambda_0))\bar{T}_{gas}(\lambda_0,\theta) \tag{11}$$

where $m_R$, $m_{O3}$, and $m_{aer}$ are the airmass for molecular scattering (Rayleigh
scattering), ozone, and aerosol, respectively, and $\tau_R$, $\tau_{O3}$, and $\tau_{aer}$ are the optical
depths for molecular scattering, ozone, and aerosol, respectively.

If the sensor output is proportional to the input energy, the following equation can be

written.

$$\bar{V}(\lambda_0) = \frac{\bar{V}_0(\lambda_0)}{R^2} \exp(-m_R(\theta)\tau_R(\lambda_0) - m_{O3}(\theta)\tau_{O3}(\lambda_0) - m_{aer}(\theta)\tau_{aer}(\lambda_0))\bar{T}_{gas}(\lambda_0,\theta)$$
$$= \frac{\bar{V}_0(\lambda_0)}{R^2} \exp(-m_T(\theta)\tau_T(\lambda_0))\bar{T}_{gas}(\lambda_0,\theta) \tag{12}$$

Here, the contribution of the diffuse radiances in the FOV is neglected.

If absorption by gas absorption lines can be ignored, eq. (12) can be written as

follows:

$$\bar{V}(\lambda_0) = \frac{\bar{V}_0(\lambda_0)}{R^2} \exp(-m_R(\theta)\tau_R(\lambda_0) - m_{O3}(\theta)\tau_{O3}(\lambda_0) - m_{aer}(\theta)\tau_{aer}(\lambda_0))$$
$$= \frac{\bar{V}_0(\lambda_0)}{R^2} \exp(-m_T(\theta)\tau_T(\lambda_0)) \tag{13}$$

Taking the logarithm of eq. (13) leads to

$\ln(\bar{V}(\lambda_0)R^2) = \ln \bar{V}_0(\lambda_0) - m_T(\theta)\tau_T(\lambda_0)$ .            (14)
If a series of measurements is taken over a range of $m_T(\theta)$ during which the optical
depth $\tau_T(\lambda_0)$ remains constant, $\bar{V}_0(\lambda_0)$ may be determined from the ordinate
intercept of a least-squares fit when one plots the left-hand side of eq. (14) versus
$m_T(\theta)$. This procedure is commonly known as the Langley-plot calibration. $\bar{V}_0(\lambda_0)$ is
the sensor output for the extra-terrestrial solar irradiance at 1 AU earth-sun distance,
and is called the calibration constant.
The Langley method which is performed assuming the same airmass of air molecule
scattering for all attenuating substances is sometimes called the normal Langley
method (Reagan et al. 1986) or the traditional Langley method (Schmid and Wehrli
1995). In this paper, "normal Langley" is used.
When the different components contributing to the attenuation have different
vertical distributions, each component has a different dependence of the airmass on
the solar zenith angle. In the refined Langley method, the contribution to the
attenuation of each component is treated separately (Thomason et al. 1983, Guzzi et al.
1985, Reagan et al. 1986, Bruegge et al. 1992, Schmid and Wehrli 1995).
The effect of the vertical distribution of ozone on the determination of the calibration
constant was examined by Thomason et al. (1983). According to their results, the
influence of the vertical distribution of ozone is large when 0.1% accuracy is required,
but at a wavelength of 500 nm, the error is at most 0.1% even if using the airmass of
the uniform mixture atmosphere.
The presence of thick stratospheric aerosol layers, such as measured immediately
after major volcanic eruptions including the Pinatubo eruption in July 1991, may
cause the airmass to be different from under ordinary conditions (Russell et al. 1993,
Dutton et al. 1994).
For the water vapor absorption band at a wavelength of 940 nm, the
Beer-Lambert-Bouguer law is not valid. In this region, the modified Langley method is
often used (Reagan et al. 1987a, Bruegge et al. 1992, Schmid and Wehrli 1995). In the
modified Langley method, the transmittance is approximated by an empirical formula.
In section 7, this modified Langley method is applied to the onsite measurement data.

## 4.2 Normal Langley method
In this section, the precision of the normal Langley is investigated,
$$\bar{V}(\lambda_0) = \frac{\bar{V}_0(\lambda_0)}{R^2} \exp(-m_R(\theta)\tau_R(\lambda_0) - m_{aer}(\theta)\tau_{aer}(\lambda_0))\bar{T}_{gas}(\lambda_0,\theta)$$
$$\approx \frac{\bar{V}_0(\lambda_0)}{R^2} \exp(-m_R(\theta)(\tau_R(\lambda_0) + \tau_{aer}(\lambda_0)))\bar{T}_{gas}(\lambda_0,\theta)$$
(15)

where $m_{aer}(\theta)$ is approximated by $m_R(\theta)$. $m_R(\theta)$ is calculated using the formula
from Kasten and Young (1989). To compute $m_{aer}(\theta)$ exactly, we would need a vertical
profile of the aerosol extinction coefficient. However, it is difficult to obtain the vertical
profile of the aerosol extinction coefficient. Therefore, $m_R(\theta)$ is often used instead of
$m_{aer}(\theta)$ (Schmid and Wehrli 1995, Holben et al. 1998).
In the case of "no gas absorption", the following equation is used:
$$\bar{V}(\lambda_0) = \frac{\bar{V}_0(\lambda_0)}{R^2}\exp(-m_T(\theta)\tau_T(\lambda_0))$$ (16)
where $m_T(\theta) = m_R(\theta)$; the same airmass is assumed for all attenuators.
Although the term for the gas line absorption is not written explicitly, if the line
absorption is in the region of the weak line limit, the absorptance (= 1 − transmittance)
is proportional to the sum of the line absorption strengths. Therefore, the
transmittance changes exponentially with the airmass.
In the case of "gas absorption", the following equation is used:
$$\bar{V}(\lambda_0) = \frac{\bar{V}_0(\lambda_0)}{R^2}\exp(-m_R(\theta)(\tau_R(\lambda_0) + \tau_{aer}(\lambda_0))\bar{T}_{gas}(\lambda_0, \theta).$$ (17)
When calculating $\bar{T}_{gas}(\lambda_0, \theta)$, the absorption of water vapor, carbon dioxide, ozone,
methane, carbon monoxide, and oxygen is only taken into consideration when the
absorptions by these gases are in the range of the response function.
It is recommended that the measurements for calibration by the Langley method be
conducted at a high mountain observatory. The MLO is one of the most suitable places
to make measurements for calibration by the Langley method. Though the air at MLO
is exceedingly transparent, it is affected in late morning and afternoon hours by
marine aerosol that reaches the observatory during the marine inversion boundary
layer breakdown under solar heating. Typically, by late morning the downslope winds
change to upslope winds, which bring moisture and aerosol-rich marine boundary
layer air up the mountainside, resulting in an abundance of orographic clouds at the
observatory (Show 1983, Perry et al. 1999). Therefore, using data taken in the morning
is recommended and used (Show 1982, Dutton et al. 1994, Holben et al 1998).
In AERONET, the variability of the determined calibration coefficient as measured
by the coefficient of variation or the relative standard deviation (CV or RSD, standard
deviation/mean) is $\sim 0.25 - 0.50\%$ for the visible and near-infrared wavelengths,
$\sim 0.5 - 2\%$ for ultraviolet and $\sim 1 - 3\%$ for the water vapor channel (Holben et al.

1998).

In this study, though using data taken in the morning is recommended, both
morning and afternoon data were used for the Langley plot. Our observation period for
calibration by the Langley method is short, about 1 month, so we want to use all the
data effectively. Furthermore, the quality of the Langley plot can be checked by an
analysis of the residuals; for acceptable data, no trend or systematic pattern is visible
when the residuals versus airmass are plotted. The residuals were carefully checked
and most results for the afternoon data were not included in the analysis.
Figure 4 shows an example of a Langley plot using the data obtained at MLO. In
these Langley plots, the data in both the morning and afternoon are plotted. The linear
regression lines were determined using the data with airmasses from 2 to 6, in the
morning. In these examples, the data in the afternoon lies close to the regression line
fitted to the morning data. On such days, the Langley plot was also applied to the
afternoon data. From these examples, by using data taken at a location with suitable
conditions, it is possible to determine a precise calibration line.
At MLO, ten to twenty measurements for the Langley calibration can usually be
taken over a period of 30 to 40 continuous observation days depending on the weather
conditions. Shortwave-infrared channels (1225, 1627, and 2200 nm) are more sensitive
to weather conditions than channels in the visible and near-infrared range, because
there are water vapor absorption bands in the shortwave-infrared channel and the
water vapor in the atmosphere tends to fluctuate.
Table 2 shows the calibration constants ($V_0$) determined using the data taken from
October 2015 to November 2015 at MLO. The calibration constants were calculated for
the following four cases.
Case 1: no gas absorption, and no temperature correction (NGABS, NTPC)
Case 2: no gas absorption, and temperature correction (NGABS, TPC)
Case 3: gas absorption, and no temperature correction (GABS, NTPC)
Case 4: gas absorption, and temperature correction (GABS, TPC)
The CV of the calibration constants ($SD/V_0$, SD is the standard deviation, $V_0$ is the
mean) were 0.2 to 1.3% except in the 940 nm channel, where the mean $V_0$ and
standard deviation were calculated from all data with weighting. The weight is
calculated from the RMS error of the regression line and the observations (see
Appendix 1). From these results, it can be seen that the calibration constant can be
reliably determined by the normal Langley method using the data taken at MLO. In
AERONET, similar results were obtained (Holben et al. 1998).
Based on the ratio of (GABS, NTPC)/(GBAS, TPC) (= (Case 3)/(Case 4)), the effect of
the temperature dependence on the 340 and 2200 nm channels was about 3 and 5%,
respectively. In the other channels, the effect of the temperature dependence is less
than 0.9%. The range of the atmospheric temperature was about 5 to 15 ºC when the
measurements for the calibration at MLO were conducted. Therefore, the effect of the
temperature dependence on the sensor output is small.
From the ratio of (NGABS, TPC)/(GABS, TPC) (= (Case 2)/(Case 4)), the effect of the
gas absorption is more than 10% in the 940 nm channel, less than 0.4% in the 1225 and
1627 nm channels, and about 1% in the 2200 nm channel. These channels have weak
gas absorption by water vapor, $CO_2$, and CO.
As seen from the ratio of (NGABS, NTPC)/(GABS, TPC) (= (Case 1)/(Case 4)), the
calibration constants, except in the 340, 940, and 2200 nm channels, can be
determined with a difference of less than 1% without consideration of the temperature
effect and gas absorption by using the data taken at MLO.
The results shown here were obtained using the data taken at MLO.
To calibrate the 940 nm channel, the vertical distribution of water vapor is necessary.
The vertical distribution of water vapor is constructed with radiosonde data from the
nearest site, precipitable water vapor (PWV) by the Global Positioning System (GPS),
and the relative humidity is measured at MLO, and the transmittance is calculated as
in Uchiyama et al. (2014). The radiosonde measurements were taken twice a day, and
the PWV by GPS were the 30-minute averages. The temporal resolution of these data
is not high enough to precisely determine the vertical distribution of the water vapor,
resulting in a large error in the calibration constant in the 940 nm channel.
Figure 5 shows the annual multiyear variation of the calibration constants ($V_0$) for
POM-02 (Calibration Reference). The lens in the visible and near-infrared region (Si
photodiode region) was replaced in 2013 and the interference filter in the 1225 nm
channel was replaced in 2014. Since insufficient data were taken due to bad weather
conditions in 2007 and 2008, the calibration could not be performed with sufficient
precision. Therefore, the degradation is not smooth in some channels.
In general, the degradation at shorter wavelengths is larger than at longer
wavelengths in the Si photodiode region. During the period from 2006 to 2012, the
changes of $V_0$ in the 340, 380, and 400 nm channels were −10% per year, −7% per year,
and −4% per year, respectively. The changes of $V_0$ in the 500, 675, and 870 nm
channels were about −1% per year, and that in the 1020 nm channel was almost zero.
These results indicate that calibration is necessary at least once a year to monitor the
degradation of $V_0$. After replacing the lens in 2013, the degradation of the 340, and
380 nm channels became smaller. The manufacturer of the sky radiometer may have
upgraded the lens.
The calibration in the shortwave-infrared channels (1225, 1627, and 220 nm) is
sensitive to weather conditions. Therefore, the interannual variation of the calibration
constants in these channels is not always smooth. However, from 2009 to 2016, the
annual change of the calibration constant in the shortwave-infrared channels was less
than 1%.

### 4.2 $V_0$ calibration transfer by direct solar measurement

The calibration constant for one instrument can be used to estimate the calibration
constant for another instrument by comparison with the simultaneous measurements
of the solar direct irradiance.
The measurements for the comparison were made every minute using the same data
acquisition system. It takes about 10 seconds to measure 11 channels at each time.
Measurements by all POM-02 are done at the same time. The calibration of time is
carried out every hour using the NTP (Network Time Protocol) Server. For data
comparison, only airmass data less than 2.5 were used on clear days. The comparisons
were made under the assumption that the filter response functions of POM-02 are the
same. When there is a difference in the filter, the relationship between the outputs of
both becomes nonlinear. When this greatly deviated from the linear relationship, the
characteristics of either filter had changed, and it is necessary to replace the filter.
Table 3 shows the results of the calibration constant transferred from POM-02
(Calibration Reference) to POM-02 (Tsukuba) and POM-02 (Fukuoka) in December
2014. The comparison measurements were conducted over 11 days for POM-02
(Tsukuba) and 8 days for POM-02 (Fukuoka). The CV ($SD/V_0$) is 0.1 to 0.5%
depending on the wavelength, where mean $V_0$ is the arithmetic mean. The CV is 0.5%
even for water vapor in the 940 nm channel; usually the fluctuation of the sensor
output is large due to fluctuations in the water vapor amount. If the weighted mean is
used as the expected value, a smaller CV than that of the arithmetic mean is expected.
The observations for the comparison depend on the weather conditions, but if there
are calibrated instruments, it is the most straightforward and accurate way to transfer
and determine the calibration constant for different instruments.
The JMA routine observation branch participated in the Fourth WMO Filter
Radiometer Comparison in Davos, Switzerland, between 28 September and 16 October
2015 (Kazadzis et al. 2018). The calibration constant of POM-02 used by them was
transferred from the POM-02 (Calibration Reference) in this study by the method
shown in this paper. In this inter-comparison campaign, the aerosol optical depths at
the 500 nm and 875 nm wavelengths were compared. The results of the comparison
showed that the JMA's POM-02 met the World Meteorological Organization (WMO)
criterion (WMO 2005). This shows that the method shown in this study is adequate.
The WMO criterion for the absolute differences of all instruments compared to the
reference is defined as follows: "95% of the measured data has to be within
0.005±0.001/$m$" (where $m$ is the airmass).

**5. Improved Langley method**
**5.1 Brief review of Improved Langley method**
In this section, the Improved Langley method is briefly reviewed.
The solar direct irradiance at the surface normal to the solar beam based on the
Beer-Lambert-Bouguer Law is written as follows:
$$F = \frac{F_0}{R^2}\exp(-m\tau)$$ (18)
where $F$ and $F_0$ are the solar irradiance at the surface and the top of the
atmosphere, respectively, $R$ is the earth-sun distance in astronomical units (AU),
$m = 1/\mu_0$ is the airmass, $\mu_0$ is the cosine of the solar zenith angle, and $\tau$ is the total
atmospheric optical depth.
The single scattered radiance by aerosol and molecules in the almucantar of the sun
is given by the following equation (Tanaka et al. 1986):

$$I_1(\mu_0,\phi) = m\tau\omega_0 P(\cos\Theta)\frac{F_0}{R^2}\exp(-m\tau)$$

$$= m\tau_{sca} P(\cos\Theta)\frac{F_0}{R^2}\exp(-m\tau)$$ (19)

where a one-layer plane-parallel atmosphere is assumed, $\tau_{sca} = \tau\omega_0$ is the layer
scattering optical depth, $\phi$ is the azimuthal angle measured from the solar principal
plane, $\omega_0$ is the single scattering albedo, and $P(\cos\Theta)$ is the normalized phase
function at the scattering angle $\Theta$. The Improved Langley method is based on these
equations.
If the sensor output is proportional to the input energy, the sensor output for the
direct solar measurement can be written as follows:
$$V = \frac{V_0}{R^2}\exp(-m\tau)$$ (20)
where $V = CF$, $V_0 = CF_0$, and $C$ is the proportional constant (sensitivity). The
contribution of scattered light in the field of view is neglected.
The sensor output for the measured single scattering $V_1$ can be written as follows:

$$V_1 = CI_1(\mu_0, \phi)\Delta\Omega$$

$$= Cm\tau\omega_0 P(\cos\Theta)\frac{F_0}{R^2}\exp(-m\tau)\Delta\Omega \tag{21}$$

$$= m\tau\omega_0 P(\cos\Theta)\frac{V_0}{R^2}\exp(-m\tau)\Delta\Omega$$

where $\Delta\Omega$ is the SVA.
From these equations, the following equations can be obtained:
$$m\tau = \frac{V_1}{\omega_0 P(\cos\Theta)\dfrac{V_0}{R^2}\exp(-m\tau)\Delta\Omega} \tag{22}$$

$$m\tau_{sca} = \frac{V_1}{P(\cos\Theta)\dfrac{V_0}{R^2}\exp(-m\tau)\Delta\Omega} . \tag{23}$$

Then from eq. (20), we get the following equations:
$$\ln VR^2 = \ln V_0 - m\tau$$
$$= \ln V_0 - m\tau_{sca}/\omega_0 \tag{24}$$

If $m$, $m\tau$, and $m\tau_{sca}$ can be obtained, the logarithm of the sensor output can be
linearly fitted with $m$, $m\tau$, and $m\tau_{sca}$. The case when the x-axis is $m$ and the y-axis
is $\ln VR^2$ corresponds to the normal Langley method, and the case when the x-axis is
$m\tau$ or $m\tau_{sca}$ and the y-axis is $\ln VR^2$ is the Improved Langley method. In the
normal Langley method, the intersection of the y-axis and the regression line is $\ln V_0$
and the slope of the regression line is $-\tau$. There are two IML methods. If the x-axis is
$m\tau$, the intersection of the y-axis and the regression line is $\ln V_0$ and the slope is $-1$.
Otherwise, if the x-axis is $m\tau_{sca}$, the intersection of the y-axis and the regression line
is $\ln V_0$ and the slope is $-1/\omega_0$. The SKYRAD package adopts the latter method.
In the SKYRAD package, two observable quantities are analyzed. One is the direct
solar irradiance (eq. (20)), and the other is defined as
$$R(\lambda,\Theta) = \frac{V(\lambda,\Theta)}{V(\lambda,0)m\Delta\Omega} \tag{25}$$

where $V(\lambda,\Theta)$ is the sensor output of the sky radiance measurement for the
scattering angle $\Theta$, $\cos\Theta = \mu_0^2 + (1-\mu_0^2)\cos\phi$, $\Delta\Omega$ is the SVA of the sky radiometer,
and $V(\lambda,0)$ is the radiometer output due to direct solar irradiance. This is the sky
radiance normalized by the direct solar irradiance.
$V(\lambda, \Theta)$ is composed of the single scattering and multiple scattering radiances.
Therefore, eq. (25) can be expressed as follows:
$$R(\lambda, \Theta) = \frac{V_1(\lambda, \Theta)}{V(\lambda, 0)m\Delta\Omega} + R_m(\lambda, \Theta)$$
$$= \tau\omega_0 P(\cos\Theta) + R_m(\lambda, \Theta)$$
(26)

where $R_m(\lambda, \Theta)$ is the contribution of multiple scattering.
In the SKYRAD package, given the initial value of the column particle volume size
distribution ($dV/d\log r$) and the complex refractive indexes, $\tau$, $P(\cos\Theta)$, and $\omega_0$,
are calculated assuming the spherical homogeneous particle. On the basis of these
single scattering properties, the multiple scattering term (second term on the right
side) in eq. (26) is evaluated, and the single scattering term (first term on the right
side) in eq. (26) can be obtained. The new $dV/d\log r$ is retrieved from the single
scattering term in eq. (26) by the inversion scheme. Using the retrieved $dV/d\log r$,
$\tau$, $P(\cos\Theta)$, and $\omega_0$ are calculated, and the observed values are reconstructed, and
then the error is calculated. Until the error satisfies the convergence condition, the
above procedure is iterated. In the above procedure, the complex refractive indexes for
each channel are fixed and the measurement data with a scattering angle of less than
30 degrees are used.
Once $m\tau$ is obtained, the calibration constants can be estimated from
$\ln V_0 = \ln VR^2 + m\tau$. However, in the SKYRAD package, $\ln V_0$ is determined from
$\ln VR^2 = \ln V_0 - m\tau_{sca}/W_0$. Comparing this equation with eq. (24), $W_0$ must be the
single scattering albedo. The single scattering albedo is defined as the ratio of the
scattering coefficient to the extinction coefficient. Therefore, the single scattering
albedo must be a value between zero and one. However, $W_0$ is frequently greater than
1. Therefore, it is treated as a constant in the estimation of $\ln V_0$. To distinguish
between $\omega_0$ and $W_0$, $W_0$ was used. The fitted error, number of measurements, and
the transmittance are checked. Then, the data passing the check criterion are chosen
as the calibration constants.

**5.2 Comparison between Improved Langley and normal Langley method**
In the Improved Langley (IML) method, the temperature dependence of the sensor
output is not usually explicitly considered. This means that the calibration constant
determined by the IML method implicitly includes the temperature dependence of the

sensor output. Before comparing the calibration constant determined by the IML method and that transferred from POM-02 (Calibration Reference), we examined how much the sensor output changes with ambient temperature change.

In Fig. 6, the monthly mean values of the inside temperature of POM-02 (Tsukuba) and the temperature of the shortwave-infrared detector are shown. As seen from the figure, these temperatures were controlled in the period from November to April. In Fig. 7, the temperature correction factors are shown, where the reference temperature is 30 ºC. In the summer, the sensor output must be corrected by 1.5 to 2% in the 340 and 380 nm channels and by 4% in the 2200 nm channel. In the other channels, the corrections were less than 0.5%: the temperature effect on these channels was small.

In Fig. 8, the calibration constants determined by the IML method from January 2014 to December 2015 are shown. To compare between the IML and normal Langley methods, the calibration constants interpolated from the calibration constants transferred from POM-02 (Calibration Reference) are also shown. The observations for the calibration transfer were conducted in December 2013, December 2014, and December 2015, and the calibration constants for POM-02 (Tsukuba) were determined. The calibration constants in other months were obtained by linear interpolation and the temperature correction factor was also taken into consideration. In Fig. 8, the running means of the monthly IML values are also shown.

For every channel, the calibration constants determined by the IML method have a seasonal variation: they are larger in the winter and smaller in the summer. The amplitude of the seasonal variation is larger than that of the temperature correction factor. Furthermore, the annual trend of the calibration constant, after removing the seasonal variation, is almost the same as the normal Langley method. Furthermore, Fig. 8 shows much higher noise of the IML method compared with calibration transfer method.

In the 380 nm channel, the calibration constant changes due to the temperature dependence of the sensor output: in the summer the calibration constant decreases by about 2%. The calibration constant ($V_0$) determined by the IML method changes by up to 6%. Even if the effect of the temperature change is subtracted from the seasonal variation, there is a difference of about 4% between the $V_0$ determined by the IML method and $V_0$ interpolated from $V_0$ determined by inter-comparison with the POM-02 (Calibration Reference). In the 400, 500, 675, and 870 nm channels, there is a

difference of 1 to 2% between the calibration coefficients, and in the 340 nm channel,
there is a difference of 3% between the calibration coefficients. In the 1020 nm channel,
since the interference filter was changed in September 2014, a direct comparison is
difficult. In Table 4, the statistics of the difference between both calibration coefficients
are shown. The RMS error is about 0.6 to 2.5% depending on the wavelength. This
result is almost the same as in Campanelli et al. (2004). However, the maximum
difference between both calibration coefficients was about 1.3 to 4.7%, and these
differences are rather large. The statistics of the 3-point running mean for the IML
method are also shown in Table 4. The errors are a bit smaller than for the
non-smoothed values: the RMS error is about 0.5 to 1.7%.

Though the period of comparison is only two years, the calibration constant by the

IML method represents the annual trend and implicitly includes the temperature
dependence of the sensor output. However, the calibration constant has a seasonal
variation of 1 to 3%, and in some cases, the maximum difference reaches about 5%. The
2% error in the calibration constant is not significant in a turbid atmosphere, but it is
significant in a clear atmosphere, such as in polar and ocean regions. Furthermore,
there is a possibility that the seasonal variation of the calibration constant causes an
artificial seasonal variation in the retrieved parameters. The seasonal variation can be
reduced by smoothing, such as with a running mean. However, over-smoothing
dampens the temperature effect of the sensor output.

For the 500 nm channel, Fig. 9 shows a scatter plot of $\Delta V_0$ and the optical depth at

500 nm, a scatter plot of $\Delta V_0$ and $W_0$, and a time series of $\Delta V_0$ from January 2014
to December 2015, where $\Delta V_0$ is the difference between $V_0$ determined by the IML
method and $V_0$ interpolated from $V_0$ determined by inter-comparison with the
POM-02 (Calibration Reference). In this case, the $V_0$ values determined by the IML
method with errors less than 0.01 were chosen, where the error is the root mean
square difference between the observations and the fitted line. As in Fig. 8, Fig. 9 (c)
shows that $\Delta V_0$ changes seasonally.

Figure 9 (a) shows that there is a negative correlation between $\Delta V_0$ and the optical

depth; the correlation coefficient is −0.31. This result is consistent with the large
amplitude of the seasonal change at short wavelengths. Since usually a shorter
wavelength corresponds to a thicker optical depth, a shorter wavelength corresponds
to a larger amplitude of seasonal change of $V_0$ by the IML method.

In Tsukuba, the aerosol optical depth is thicker in the summer and thinner in the

winter. Therefore, the seasonal change of $V_0$ by the IML method seems to be related
to the optical thickness. However, Fig. 9 (b) also shows that $\Delta V_0$ and $W_0$ are
negatively correlated, specifically having a correlation coefficient of −0.59, and that
even if the correct $W_0$ is determined, the $\Delta V_0$ are scattered with a width of about
$1.0 \times 10^{-5}$. Since $W_0$ is a parameter related to the single scattering albedo or refractive
index, this indicates that the error depends not only on the optical depth but also on
the refractive index. There is a possibility that the seasonal variation of $V_0$ by the IML
method may also be related to the seasonal variation of the refractive index.

In the current Improved Langley method, the refractive index is fixed. We used (1.5,
−0.001) for all wavelengths as the initial value of the refractive index when using the
SKYRAD package. However, this value may not be appropriate, and the further
development of the method to determine $V_0$ while changing the refractive index is a
topic for future work.

### 6. Calibration using the calibrated light source

In this section, the accuracy of the calibration using the calibrated integrating
sphere is described. If POM-02 can be calibrated using the calibrated light source, then
POM-02 can be calibrated quickly without being influenced by the weather.

In this study, the integrating sphere, which is calibrated and maintained by the
Japanese Aerospace Exploration Agency (JAXA), was used (Yamamoto et al. 2002).
This integrating sphere is used to calibrate the radiometers which are used to validate
satellite remote sensing products.

To use the light source, the extra-terrestrial solar irradiance and the SVA and
spectral response function of the sky radiometer are necessary, as well as the radiance
emitted by the light source. The extra-terrestrial solar irradiance by Gueymard (2004)
was used here, along with the SVA obtained by processing the solar disk scan data.

When the integrating sphere is measured by POM-02, the sensor output is written
as follows:
$$V_{sph}(\lambda_0) = \int_{\Delta\lambda} C(\lambda)\varphi(\lambda)I_{sph}(\lambda)d\lambda \cdot \Delta\Omega \bigg/ \int_{\Delta\lambda} \varphi(\lambda)d\lambda \qquad (27)$$
where $V_{sph}(\lambda_0)$ is the sensor output in channel $\lambda_0$, $C(\lambda)$ is the sensitivity at
wavelength $\lambda$, $\varphi(\lambda)$ is the spectral response function of the interference filter,
$I_{sph}(\lambda)$ is the spectral radiance from the integrating sphere at wavelength $\lambda$, and the
emitted radiance from the integrating sphere is assumed to be homogeneous. This
equation is approximated as follows:
$$V_{sph}(\lambda_0) \cong C(\lambda_0)\overline{I}_{sph}(\lambda_0) \cdot \Delta\Omega \qquad (28)$$
where
$$\overline{I}_{sph}(\lambda_0) = \int_{\Delta\lambda} \varphi(\lambda)I_{sph}(\lambda)d\lambda \bigg/ \int_{\Delta\lambda} \varphi(\lambda)d\lambda . \qquad (29)$$
When the extra-terrestrial solar irradiance is measured, the sensor output is written
as follows:
$$V_{sun}(\lambda_0) = \int_{\Delta\lambda} C(\lambda)\varphi(\lambda)F_0(\lambda)d\lambda \bigg/ \int_{\Delta\lambda} \varphi(\lambda)d\lambda \qquad (30)$$
where $V_{sun}(\lambda_0)$ is the sensor output in channel $\lambda_0$, and $F_0(\lambda)$ is the
extra-terrestrial solar spectral irradiance at 1 AU. This equation is approximated as
follows:
$$V_{sun}(\lambda_0) \cong C(\lambda_0)\overline{F}_0(\lambda_0) \qquad (31)$$
where
$$\overline{F}_0(\lambda_0) = \int_{\Delta\lambda} \varphi(\lambda)F_0(\lambda)d\lambda \bigg/ \int_{\Delta\lambda} \varphi(\lambda)d\lambda . \qquad (32)$$
From eqs. (28) and (31), $V_{sun}(\lambda_0)$ is written as follows:
$$V_{sun}(\lambda_0) \cong V_{sph}(\lambda_0)\frac{\overline{F}_0(\lambda_0)}{\overline{I}_{sph}(\lambda_0) \cdot \Delta\Omega} . \qquad (33)$$
In Table 5, the calibration constants for POM-02 (Calibration Reference) determined
from the integrating sphere measurement are compared with the results of the
Langley method. At POM-02 (Calibration Reference), the relative difference was 0.7 to
7.6% in channels 2 to 8 (380 to 1020 nm), and 0.5 to 1.8% in channels 9 to 11 (1225,
1627, and 2200 nm). The integrating sphere used in channels 2 to 8 is different from
that in channels 9 to 11.
The value of the extra-terrestrial solar spectrum is dependent on the database. In
Fig. 10, four data sets are shown (Thuillier et al. 2003, Gueymard 2004, and 1985
Wehrli Standard Extraterrestrial Solar Irradiance Spectrum (Wehrli 1985, Neckel and
Labs 1981), Chance and Kurucz 2010). The value is a mean value weighted by the
response function of a triangle with full width at half maximum (FWHM) of 10 nm.
The ratios of the solar spectrum to Gueymard (2004) are also shown. These figures
show that there is a several percent difference in the values depending on the
wavelength. The SVA uncertainty is 1% (see Part II); the disk scan data were taken at
MLO, where measurement conditions were good for the solar disk scan. The
uncertainty of the integrating sphere was 1.7% (Yamamoto et al. 2002). Considering
the magnitude of these errors, the above differences in the calibration constants seem
reasonable. However, to reduce the optical depth error below 0.01, a calibration
coefficient error of several percent is too large. For estimating the optical depth from
measurements of the direct solar irradiance, the calibration coefficient determined by
the Langley method is better. These issues were also pointed out by Shaw (1976), and
Schmid and Wehrli (1980). The calibration using the standard lamp remains
unchanged.

### 717    7. Calibration of 940 nm channel

The calibration constant depends on the extra-terrestrial solar irradiance in the 940
nm band, the spectral response function of the interference filter, the spectral
sensitivity of the detector, and the transmittance of radiometer optics. Calibration
methods for the 940 nm channel, which is in the water vapor absorption band, have
been considered extensively in previous studies (Reagan et al. 1987a, 1987b, 1995;
Bruegge et al. 1992; Thome et al. 1992, 1994; Michalsky et al. 1995, 2001; Schmid et al.
1996, 2001; Shiobara et al. 1996; Halthore et al. 1997; Cachorro et al. 1998;
Plana-Fattori et al. 1998, 2004; Ingold et al. 2000; Kiedron et al. 2001, 2003). For
example, Uchiyama et al. (2014) developed the Langley method which takes into
account the gas absorption, and the empirical relationship between the transmittance
and precipitable water vapor (PWV) was determined from the theoretical calculation
using the spectral response function and the model atmosphere. The PWV is estimated
from the transmittance for the 940 nm channel. The empirical formula is usually used
for the transmittance of the 940 nm channel by water vapor.
Most POM-02 users have taken measurements without calibrating the 940 nm
channel over a long time. To make use of these accumulated data, it is necessary to
develop a calibration method using data at the observation site. Campanelli et al.
(2014) developed a method to determine the calibration constant and parameters for
the empirical formula of the transmittance using the on-site surface meteorological
data and simultaneous POM-02 data. However, it is difficult to obtain the empirical
formula for transmittance by the column water vapor from the surface measurement
data.
In this study, given the spectral response function, the empirical transmittance
formula is produced by the method shown in Uchiyama et al. (2014). Then, the
modified Langley method shown below is performed using the empirical formula and
the observation data.
The water vapor transmittance is approximated as follows:

$Tr(\mathrm{H2O}) = \exp(-a(m \cdot pwv)^b)$                                           (34)

where $a$ and $b$ are fitting coefficients (see Appendix 2), and $pwv$ is PWV.
The sensor output $V$ is written as follows (Uchiyama et al. 2014):
$$V = \frac{V_0}{R^2} \exp(-m(\tau_{aer} + \tau_R)) Tr(\mathrm{H2O})$$
$$= \frac{V_0}{R^2} \exp(-m(\tau_{aer} + \tau_R)) \exp(-a(m \cdot pwv)^b)$$
                     (35)

where $V_0$ is the calibration coefficient, $R$ is the distance between the earth and the
sun, $\tau_{aer}$ is the aerosol optical depth at 940 nm, and $\tau_R$ is the optical depth of the
molecular scattering (Rayleigh scattering). The aerosol optical depth $\tau_{aer}$ at 940 nm is
interpolated from the optical depth at 870 and 1020 nm. When interpolating $\tau_{aer}$ at
940 nm, $\tau_{aer}$ was assumed to be proportional to $\lambda^{-\alpha}$, where $\lambda$ is the wavelength.
The above equation can be rewritten as follows:
$\ln VR^2 + m(\tau_{aer} + \tau_R) = \ln V_0 - a(pwv)^b m^b$.                           (36)
The parameters on the left-hand side are known: $V$ is the measurement value, $R$
and $m$ can be calculated from the solar zenith angle, and $\tau_R$ is estimated from the
surface pressure. For example, $R$ can be calculated with the simplified formula in
Nagasawa (1981), $m$ can be calculated as in Kasten and Young (1989), and $\tau_R$ can
be calculated as in Asano et al. (1983). In the case of POM-02, the sensor output is
current, and the unit of the measurement value $V$ is Ampere (A). If $pwv$ is constant,
then the right-hand side of the equation is a linear function of $m^b$. Therefore, the
values on the left-hand side can be fitted by a linear function of $m^b$, and the
intersection of the y-axis and the fitted line is $\ln V_0$.
Before the above-mentioned method was applied to the MRI data, it was first applied
to the data taken at MLO, which has more stable weather conditions than Tsukuba,
MRI. The results applied to the data taken at MLO in October and November 2014 and
in October and November 2015 are shown in Table 6.
The calibration coefficients determined in 2014 and 2015 were $2.2973 \times 10^{-4}$ A
($SD/V_0 = 0.052$) and $2.2954 \times 10^{-4}$ A ($SD/V_0 = 0.047$), respectively.
The calibration coefficients determined by the Langley method with consideration of
gas absorption in 2014 and 2015 were $2.3364 \times 10^{-4}$ A ($SD/V_0 = 0.093$) and $2.3157 \times 10^{-4}$
A ($SD/V_0 = 0.097$), respectively. Though the difference in the calibration coefficient
between the Langley method with consideration of the gas absorption and the modified
Langley method is 1.7% in 2014 and 0.9% in 2015, these calibration coefficients are
very similar. The CV of the modified Langley method is smaller than the method which
takes account of gas absorption more precisely than the modified Langley method. This
may be due to errors in the estimates of the water vapor amount and distribution: the
PWV is obtained from the GPS PWV, which has a low time resolution (30-minute
average) and some data are missing, and the vertical distribution is estimated from
only two radiosonde measurements per day near MLO.

The water vapor amount tends to fluctuate. Though the restriction that the PWV be

constant is severe, the above method is applied to the data taken at Tsukuba, MRI and
the calibration constants are compared with the calibration constant for POM-02
(Calibration Reference), which was calibrated by the Langley method with
consideration of the gas absorption using the data taken at MLO and interpolated to
the observation day (see Table 7).

The ratio of the calibration coefficients in the period from December 14, 2014 to

January 5, 2015 (10 cases) was 1.0094, and in the period from December 1, 2015 to
December 30, 2015 (17 cases) it was 0.99818. Thus, the difference between the two
methods is less than 1%.

Although it seems that the above-mentioned modified Langley method does not work

well at all locations and under all weather conditions, the calibration constant of the
940 nm channel could be determined by applying the above-mentioned method on a
suitable stable and fine day at the observation site. We applied Langley method to data
in the airmass range between 2 and 6. Therefore, a stable interval of 1 to 2 hours is
necessary. The quality of the Langley plot can be checked by an analysis of the
residuals; for acceptable data, no trend or systematic pattern is visible when the
residuals versus airmass are plotted. The 940 nm channels at many observation sites
have not been calibrated and are not used. The application of the modified Langley
method to the on-site observation data is the next best solution.

**8. Calibration coefficients of shortwave-infrared channels**

The measurements for the shortwave-infrared channels, 1225, 1627, and 2200 nm, of

POM-02 have been performed at many SKYNET sites, but the data have not been
analyzed, because most POM-02 users cannot calibrate these channels by themselves.

These channels can be calibrated with the Langley method with a reasonable

precision by taking into account the gas absorption. However, many users cannot make
these measurements for the Langley method. Furthermore, the scattering of light in
these channels is small and the IML method cannot be applied.
For some observation days, data with a very high correlation between channels may
be obtained. In this case, if the calibration constant of one channel is known, then the
calibration constants of the other channels can be inferred. The general method for the
case when the ratio of the optical depths is constant was shown by Forgan (1994).
In this study, by assuming that the channels in the visible and near-infrared region
including the 940 nm channel are calibrated, a similar method was applied to the
shortwave-infrared channels to determine the calibration constant and the precision
was investigated.
The sensor output of POM-02 is written as follows:
$$V = \frac{V_0}{R^2} \exp(-m(\tau_{aer} + \tau_R))Tr(gas) \tag{37}$$
where $V$ is the sensor output, $V_0$ is the calibration constant, $R$ is the distance
between the earth and the sun, $m$ is the airmass, $\tau_{aer}$ is the aerosol optical depth,
$\tau_R$ is the optical depth of the molecular scattering (Rayleigh scattering), and $Tr(gas)$
is the transmittance of the gas absorption.
The sensor output for channels 1 and 2 are as follows:
$$V_1 = \frac{V_{01}}{R^2} \exp(-m(\tau_1 + \tau_{R1}))Tr_1(gas) \tag{38}$$
$$V_2 = \frac{V_{02}}{R^2} \exp(-m(\tau_2 + \tau_{R2}))Tr_2(gas). \tag{39}$$
The calibration constant of channel 1 is assumed to be known and that of channel 2 is
determined.
From eqs. (38) and (39), the following equation is obtained:
$$\frac{V_2}{V_1} = \frac{V_{02} \exp(-m(\tau_2 + \tau_{R2}))Tr_2(gas)}{V_{01} \exp(-m(\tau_1 + \tau_{R1}))Tr_1(gas)}.$$
Therefore,
$$\ln\frac{V_2}{V_1} + m(\tau_{R2} - \tau_{R1}) - \ln\frac{Tr_2(gas)}{Tr_1(gas)} = \ln\frac{V_{02}}{V_{01}} - m(\tau_2 - \tau_1)$$
$$= \ln\frac{V_{02}}{V_{01}} - (\frac{\tau_2}{\tau_1} - 1)\tau_1 m \tag{40}$$

If the water vapor amount is estimated from the 940 nm channel, and the mixing
ratio of $CO_2$ and CO is given, then the transmittance of gas can be estimated. Given
the observation time and the latitude and longitude of the observation site, the
airmass is calculated, and $\tau_{R1}$ and $\tau_{R2}$ are calculated from the surface pressure.
Therefore, the left-hand side of eq. (40) is known. Furthermore, if the ratio of the
optical depth $\tau_2/\tau_1$ is constant, then this equation is a linear function of $m\tau_1$.
Therefore, the intersection of the y-axis and the linearly fitted line is $\ln V_{02}/V_{01}$, and if
$V_{01}$ is known, then $V_{02}$ is also known. Although this condition is not always satisfied,
sometimes a linear fit will provide sufficient accuracy.
This method was applied to the data of POM-02 (Calibration Reference) from
December 2014 to December 2015. The 500 nm was chosen as channel 1 in eq. (40).
The data used here had an RMS error of 0.005. In Fig. 11(a), the monthly mean of
$V_{02}/V_{01}$ and the standard deviation are shown. The lines of the ratio, which are
interpolated from the calibration constant determined using the data taken in October
and November of 2014 and 2015 at MLO, are also shown. In Fig. 11 (b), the ratio of the
calibration constant by the above method and the interpolated value of the calibration
constant determined from MLO data are shown. In the 1627 nm channel, the
differences are less than 2% throughout the year and the differences in December and
January are less than 1%. In the 1225 nm channel, the differences are less than 2%
except in April 2015. In the 2200 nm channel, the differences in some months are more
than 3%. However, in December 2015, the differences in all channels are less than 1%,
0.8, 0.4, and 0.1%, respectively. This shows that the difference between the calibration
constant determined by the method shown here and that determined by the Langley
method is less than 1% under suitable conditions. Currently, there is no method to
calibrate the shortwave-infrared channel from on-site observation data. The method
shown here is the next best solution.

**9. Summary and conclusion**
Atmospheric aerosols are an important constituent of the atmosphere. Measurement
networks covering an extensive area from ground and space have been developed to
determine the spatiotemporal distribution of aerosols. SKYNET is a ground-based
monitoring system using sky radiometers POM-01 and POM-02 manufactured by
Prede Co. Ltd., Japan. To improve their measurement precision, it is important to
know the characteristics of the instruments and precisely calibrate them accordingly.
There are two constants that we must determine to make accurate measurements.
One is the calibration constant, and the other is the SVA of the radiometer. The
calibration constant is the output of the radiometer to the extra-terrestrial solar
irradiance at the mean earth-sun distance (1 AU) at the reference temperature.
Additionally, the temperature dependence of the sensor output is another important
characteristic.
In this study, the data obtained by two sky radiometers POM-02 of the JMA/MRI are
considered. One of the sky radiometers is used as a calibration reference, and the other
is used for continuous measurement at the Tsukuba MRI observation site.
The sensor output of POM-02 is dependent on the environmental temperature. The
temperature dependence of the sensor output in the 340, 380, and 2200 nm channels
was larger than in other channels. For example, the sensor output in the 340 and 380
nm channels of POM-02 (Tsukuba) increased at a rate of about 1.5% per 10 degrees,
and that in the 2200 nm channel increased at a rate of about 3% per 10 degrees. In the
other channels, the sensor output increased at a rate of less than 1% when the sensor's
internal temperature was 0 to 40 ºC. The temperature dependence of the two POM-02
examined here was different for each instrument. If we want to make accurate
measurements, we need to measure the temperature dependence for each instrument
or use the channels with a small temperature dependence.
For the measurement at Tsukuba, the temperature inside the POM-02 (Tsukuba)
was controlled during the winter and spring seasons from November to April, but was
not regulated, and thus was high during the summer. In the summer, sensor output
must be corrected by 1.5 to 2% in the 340 and 380 nm channels and by 4% in the 2200
nm channel. In the other channels, the corrections were less than 0.5%.
As well as determining the precision of the IML method, this study investigated the
precision of the normal Langley method (i.e., the same airmass of air molecule
scattering for all attenuating substances) and of the calibration transfer. From the
data taken at MLO, the CV in the calibration constants determined by the normal
Langley method $(\mathrm{SD}/V_0)$ was 0.2 to 1.3%, except in the 940 nm channel. The effect of
gas absorption was more than 10% in the 940 nm channel, but was less than 0.4% in
the 1225 and 1627 nm channels and less than 1% in the 2200 nm channel, which all
have weak gas absorption.
The comparison measurements for transferring the calibration constant were
conducted in December at Tsukuba over about ten days. The CV $(\mathrm{SD}/V_0)$ for the
transfer method was 0.1 to 0.5% depending on the wavelength. Though the
measurements for the comparison depend on the weather conditions, if there are
calibrated instruments, then it is a straightforward and accurate way to determine the
calibration constant.
The long-term changes in the calibration constants $(V_0)$ for POM-02 (Calibration
Reference) were also investigated. Roughly speaking, the degradation in the shorter
wavelengths was larger than that in the longer wavelengths in the Si photodiode
region. The changes in the 340 nm channel were −10% per year from 2006 to 2012.
After replacing the lens in 2013, the degradation of the 340 and 380 nm channels

became smaller. The manufacturer of the sky radiometer may have upgraded the lens. The change in the shortwave-infrared region (thermoelectrically cooled InGaAs photodiode) was less than 1% from 2009 to 2016. These results indicate that calibration of the instruments is necessary at least once a year to monitor the degradation of $V_0$.

The calibration constant determined by the IML method and that transferred from the POM-02 (Calibration Reference) were compared using the data taken at Tsukuba from December 2013 to December 2015.

For every channel, the calibration constants determined by the IML method had a seasonal variation of 1 to 3%. The calibration constants determined by the IML method implicitly include the temperature dependence of the sensor output. However, even if the change due to the temperature variation is subtracted from the seasonal variation, there is a difference of 1 to 4% between the two calibration coefficients. The RMS errors of the differences between the two calibration coefficients were about 0.6 to 2.5%. This result is almost the same as that of Campanelli et al. (2004). However, in some cases, the maximum difference reached up to 5%. Furthermore, the annual trend of the calibration constant excluding the seasonal variation was almost the same as for the normal Langley method. Furthermore, the calibration constants determined by the IML method had much higher noise than those transferred from the reference.

In order to investigate the error characteristics of the IML method, the relationship between $\Delta V_0$ and the optical depth and the relationship between $\Delta V_0$ and $W_0$ were investigated. $\Delta V_0$ is the difference between $V_0$ determined by the IML method and $V_0$ interpolated from $V_0$ determined by inter-comparison with the reference POM-02. As a result, it was found that $\Delta V_0$ and the optical depth were correlated. In Tsukuba, the aerosol optical depth changes seasonally. Therefore, the seasonal change of $V_0$ by the IML method seems to be related to the optical depth. Furthermore, $\Delta V_0$ and $W_0$, which is related to single scattering albedo or refractive index, were also correlated. In the current IML method, the refractive index is fixed. It is necessary to develop the proposed method to determine $V_0$ while changing the refractive index in the future.

We also tried to determine $V_0$ using the calibrated integrating sphere as the light source. The relative differences of $V_0$ were about 1 to 8% depending on the wavelength. Considering the magnitude of the errors in the extra-terrestrial solar spectrum, SVA, and the integrating sphere, the above differences in the calibration constants seem reasonable. However, to reduce the optical depth error below 0.01, an error of several percent in the calibration coefficient is too large.

The calibration method for water vapor in the 940 nm channel was considered using

the on-site measurement data. $V_0$ was determined by the modified Langley method
using a pre-determined empirical transmittance equation. The differences in the
calibration coefficients between the normal Langley method and the modified Langley
method were less than 1% on suitable stable and fine days.

The calibration method for the shortwave-infrared 1225, 1627, and 2200 nm

channels was also considered using the on-site measurement data. It is assumed that
channels in the visible and near-infrared wavelength region and the 940 nm channel
are calibrated. Then, if the ratio of the optical depths between two channels is constant,
the logarithm of the ratio of the sensor output can be written as a linear function of the
airmass. Here, the calibration constant for one of the two channels is known and the
transmittance of water vapor is calculated using the PWV estimated from the 940 nm
channel. By fitting the logarithm of the ratio of sensor output to a linear function of the
airmass, the ratio of the calibration constants is determined. By this method, the
calibration constants could be determined within a 1% difference from the value by the
Langley method on suitable days with good weather conditions.

In this study, it is shown that some channels have a non-negligible temperature

dependence in the sensor output and that the calibration constants determined by the
IML method showed a seasonal variation. In channel 2 (380 nm), the maximum error
reached about 5%. Reducing the uncertainty of the IML method is a task for future
work, along with the problems related to the determination of calibration constants. In
particular, the calibration constants for the 940 nm channel and the
shortwave-infrared channels must be determined using on-site measurement data.

**Appendix** 1
Weighted mean of calibration constant

Let $\sigma$ be the uncertainty of $V_0$.


$$\ln(V_0 \pm \sigma) = \ln V_0 (1 \pm \frac{\sigma}{V_0})$$

$$= \ln V_0 + \ln(1 \pm \frac{\sigma}{V_0})$$

$$\simeq \ln V_0 \pm \frac{\sigma}{V_0}$$

where $\sigma/V_0 \ll 1$.
Therefore, the uncertainty of $\ln V_0$ is $\sigma/V_0$.
Let us use the root mean square $(= \sigma_L)$ of the residual from the linear regression line
of the Langley plot as the uncertainty of $\ln V_0$.

$$\sigma_L = \frac{\sigma}{V_0}$$
Therefore, the uncertainty of $V_0$ is $\sigma = \sigma_L V_0$.
The weighted mean and standard deviation of $V_0$ were calculated by weighting
$$1 / \sigma^2 = 1 / (\sigma_L V_0)^2.$$

**Appendix 2**
Coefficients of water vapor transmittance.
Details of the method for determining the coefficients $a$ and $b$ are described in
Uchiyama et al. (2014). The coefficients $a$ and $b$ depend on the vertical structure of
the atmospheric temperature and humidity. Therefore, it is difficult to choose suitable
values that can be applied under all atmospheric conditions. The range of variability of
transmittance for an atmospheric profile is limited. Atmospheric transmittance is
computed for a broad range of atmospheric conditions, and values for $a$ and $b$ were
chosen that best fit the ensemble conditions.
The value of coefficients determined by our method for POM-02 (Calibration
Reference) are $a = 0.139186$, $b = 0.631$. The values of the coefficients for the
trapezoidal spectral response function, which has full width at a half maximum of 10
nm and central wavelength of 940 nm, are $a = 0.147101$, $b = 0.625$.

**Acknowledgements**
This work was supported by the NIES GOSAT-2 project, Japan. This work was
partially supported by JSPS KAKENHI Grant Number JP17K00531. The authors
would like to thank Dr. Forgan and two anonymous reviewers for their useful
comments.

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

Table titles
Table 1 Nominal specifications of the response functions of POM-02.

Table 2 Example of calibration constants ($V_0$) determined from the data taken at MLO.

Table 3 Results of the calibration constant transferred from POM-02 (Calibration
Reference) to POM-02(Tsukuba) and POM-02 (Fukuoka) in December 2014.

Table 4 Statistics of difference between the IML method and normal Langley method.

Table 5 Calibration constants for POM-02 determined by using the calibrated
integrating sphere measurement.

Table 6 Calibration constant at 940 nm by the modified Langley method using the data
taken at MLO.

Table 7 Same as Table 6 but using the data taken at Tsukuba, MRI.


Figure captions
Fig. 1 Relation between the inside temperatures of the instrument and the ambient
environmental temperature for POM-02 (Calibration Reference).

Fig. 2 Relation between the sensor output and the inside temperature near the filter
wheel for POM-02 (Calibration Reference). The sensor output is normalized by that at
20 °C. The error bars are the standard deviation. (a) 340, 380, 400, and 500 nm. (b) 675,
870, 940, and 1020 nm. (c) 1225, 1627, and 2200 nm.

Fig. 3 Same as Fig. 2 but for POM-02 (Tsukuba).

Fig. 4 Examples of Langley plots using the data obtained at MLO, on November 3,
2015. The sensor output of POM-02 is current: the unit is Ampere (A).

Fig. 5 Annual variation of the calibration constants ($V_0$) for POM-02 (Calibration
Reference). The sensor output of POM-02 is current. The unit of $V_0$ is Ampere (A).

Fig. 6 Monthly mean values and standard deviation of the inside temperature of
POM-02 (Tsukuba) (blue line) and the temperature of the shortwave-infrared detector
(red line) from December 2013 to December 2016.

Fig. 7 Monthly means of the temperature correction factors and standard deviation for
POM-02 (Tsukuba) from December 2013 to December 2016.

Fig. 8 Time series of the calibration constant for POM-02 (Tsukuba) from January 2014
to December 2015. Blue open squares with error bars denote the calibration constants
determined by the IML method. The green line shows the 3-point running mean of IML,
and the red line is the calibration constant interpolated from calibration constants
transferred from POM-02 (Calibration Reference). The unit of $V_0$ is Ampere (A). A
double-headed arrow shows 2% width.

Fig. 9 (a) Scatter plot of $\Delta V_0$ for the 500 nm channel and the optical depth at 500 nm.
(b) Scatter plot of $\Delta V_0$ and $W_0$ for the 500 nm channel. (c) Time series of $\Delta V_0$ for the
500 nm channel from January 2014 to December 2015. $\Delta V_0$ is the difference between
$V_0$ determined by the IML method and $V_0$ interpolated from $V_0$ determined by
inter-comparison with POM-02 (Calibration Reference). The unit of $V_0$ and $\Delta V_0$ is
Ampere (A).

Fig. 10 (a) Extra-terrestrial solar spectra. The value is a mean value weighted by the
response function of a triangle with FWHM of 10 nm. The red line is Gueymard (2004),
the blue line is Thuillier et al. (2003), the green line is Wehrli (1985), and black line is
Chance and Kurucz (2010). (b) Ratios of the solar spectrum to Gueymard (2004).

Fig. 11 (a) Monthly mean of $V_{02}/V_{01}$ and the standard deviation; here
$V_{01} = V_0(500 \text{ nm})$. (b) Ratio of $V_{02}$ to the interpolated value of the calibration constant
determined by the Langley method. The red symbols are 1225 nm, blue ones are 1627
nm, and green ones are 2200 nm.

Table 1    Nominal filter specification.

| Channel No. | Wavelength (nm) | FWHM (nm) | Max. Transmittance | Blocking | Blocking wavelength | Detector |
|---|---|---|---|---|---|---|
| − | 315(±0.6)* | 3.0(±0.6) | >30% | $1.0 \times 10^{-5}$ | 200 – 1200 nm | Si photodiode |
| 1 | 340(±0.6) | 3.0(±0.6) | >30% | $1.0 \times 10^{-5}$ | 200 – 1200 nm | Si photodiode |
| 2 | 380(±0.6) | 3.0(±0.6) | >30% | $1.0 \times 10^{-5}$ | 200 – 1200 nm | Si photodiode |
| 3 | 400(±0.6) | 10.0(±2.0) | >30% | $1.0 \times 10^{-5}$ | 200 – 1200 nm | Si photodiode |
| 4 | 500(±2.0) | 10.0(±2.0) | >30% | $1.0 \times 10^{-5}$ | 200 – 1200 nm | Si photodiode |
| 5 | 675(±2.0) | 10.0(±2.0) | >30% | $1.0 \times 10^{-5}$ | 200 – 1200 nm | Si photodiode |
| 6 | 870(±2.0) | 10.0(±2.0) | >30% | $1.0 \times 10^{-5}$ | 200 – 1200 nm | Si photodiode |
| 7 | 940(±2.0) | 10.0(±2.0) | >30% | $1.0 \times 10^{-5}$ | 200 – 1200 nm | Si photodiode |
| 8 | 1020(±2.0) | 10.0(±2.0) | >30% | $1.0 \times 10^{-5}$ | 200 – 3000 nm | Si photodiode |
| 9 | 1225(±2.0)** | 20.0(±2.0) | >30% | $1.0 \times 10^{-5}$ | 600 – 3000 nm | InGaAs photodiode |
| 10 | 1627(±2.0) | 20.0(±2.0) | >30% | $1.0 \times 10^{-5}$ | 600 – 3000 nm | InGaAs photodiode |
| 11 | 2200(±2.0) | 20.0(±2.0) | >30% | $1.0 \times 10^{-5}$ | 600 – 3000 nm | InGaAs photodiode |

FWHM: Full Width at Half Maximum

*: 315 nm channel is not used by JMA/MRI.

**: 1225 nm channel is used by JMA/ MRI.



Table 2   Example of calibration constants ($V_0$) determined by using the data taken at MLO.

| | Wavelength (nm) | 340 | 380 | 400 | 500 | 675 | 870 | 940 | 1020 | 1225 | 1627 | 2200 |
|---|---|---|---|---|---|---|---|---|---|---|---|---|
| Case 1 | V0 (NGABS,NTPC)($\times10^{-4}$) | 0.19885 | 0.39332 | 1.6412 | 2.7745 | 3.2852 | 2.4791 | 1.9936 | 1.5513 | 0.88278 | 1.4410 | 0.72407 |
| | SD($\times10^{-4}$) | 0.00112 | 0.00166 | 0.0059 | 0.0082 | 0.0058 | 0.0146 | 0.1620 | 0.0080 | 0.00889 | 0.0086 | 0.00411 |
| | CV(=SD/V0) | 0.00564 | 0.00421 | 0.0036 | 0.0030 | 0.0018 | 0.0059 | 0.0812 | 0.0052 | 0.01007 | 0.0060 | 0.00567 |
| Case 2 | V0 (NGABS,TPC)($\times10^{-4}$) | 0.20470 | 0.39515 | 1.6473 | 2.7638 | 3.2581 | 2.4825 | 1.9814 | 1.5643 | 0.87554 | 1.4287 | 0.68906 |
| | SD($\times10^{-4}$) | 0.00187 | 0.00160 | 0.0058 | 0.0079 | 0.0068 | 0.0144 | 0.1612 | 0.0094 | 0.00928 | 0.0090 | 0.00860 |
| | CV(=SD/V0) | 0.00915 | 0.00406 | 0.0035 | 0.0028 | 0.0021 | 0.0058 | 0.0814 | 0.0060 | 0.01060 | 0.0063 | 0.01248 |
| Case 3 | V0 (GABS,NTPC) ($\times10^{-4}$) | 0.19885 | 0.39331 | 1.6412 | 2.7746 | 3.2852 | 2.4791 | 2.3105 | 1.5516 | 0.88512 | 1.4422 | 0.73047 |
| | SD($\times10^{-4}$) | 0.00112 | 0.00165 | 0.0059 | 0.0082 | 0.0058 | 0.0146 | 0.2119 | 0.0080 | 0.00822 | 0.0084 | 0.00428 |
| | CV(=SD/V0) | 0.00564 | 0.00420 | 0.0036 | 0.0029 | 0.0018 | 0.0059 | 0.0917 | 0.0051 | 0.00928 | 0.0058 | 0.00587 |
| Case 4 | V0 (GABS,TPC) ($\times10^{-4}$) | 0.20469 | 0.39516 | 1.6473 | 2.7640 | 3.2582 | 2.4825 | 2.2968 | 1.5651 | 0.87843 | 1.4300 | 0.69666 |
| | SD($\times10^{-4}$) | 0.00188 | 0.00161 | 0.0058 | 0.0078 | 0.0068 | 0.0144 | 0.2092 | 0.0097 | 0.00861 | 0.0090 | 0.00872 |
| | CV(=SD/V0) | 0.00917 | 0.00407 | 0.0035 | 0.0028 | 0.0021 | 0.0058 | 0.0911 | 0.0062 | 0.00980 | 0.0063 | 0.01251 |
| | No. of data | 22 | 22 | 22 | 22 | 22 | 22 | 22 | 22 | 22 | 22 | 22 |
| | (Case 1)/(Case 4) − 1.0 | −0.0285 | −0.0047 | −0.0037 | 0.0038 | 0.0083 | −0.0014 | −0.1320 | −0.0088 | 0.0050 | 0.0077 | 0.0393 |
| | (Case 2)/(Case 4) − 1.0 | 0.0000 | 0.0000 | 0.0000 | −0.0001 | 0.0000 | 0.0000 | −0.1373 | −0.0005 | −0.0033 | −0.0009 | −0.0109 |
| | (Case 3)/(Case 4) − 1.0 | −0.0285 | −0.0047 | −0.0037 | 0.0038 | 0.0083 | −0.0014 | 0.0060 | −0.0086 | 0.0076 | 0.0085 | 0.0485 |

⬛ : ABS(ERR)> 0.03          ⬜ : ABS(ERR)<0.01

V0: mean value of calibration constant in 2015 MLO observation (the unit of V0 is Ampere(A))

SD: standard deviation

CV: coefficient of variation or relative standard deviation (=SD/V0)

GABS: consideration of gas absorption

NGABS: no consideration of   gas absorption

TPC: consideration of temperature correction

NTPC: no consideration of temperature correction

Table 3 Results of the calibration constant transferred from POM-02 (Calibration Reference) to POM-02(Tsukuba) and POM-02 (Fukuoka) in December 2014.

| Site | Tsukuba | | | | | | | | | | |
|---|---|---|---|---|---|---|---|---|---|---|---|
| Period | 2015/12/01 to 2016/01/01 | | | | | | | | | | |
| No. of days | 11 | | | | | | | | | | |
| SN | PS1202091 | Calibrated by Sky radiometer PS1207831 | | | | | | | | | |

| Wavelength (nm) | 340 | 380 | 400 | 500 | 675 | 870 | 940 | 1020 | 1225 | 1627 | 2200 |
|---|---|---|---|---|---|---|---|---|---|---|---|
| $V_0$ (×10⁻⁴) | 0.17469 | 0.25711 | 1.1621 | 2.9248 | 3.4792 | 2.2969 | 1.9900 | 0.79227 | 0.87065 | 1.4074 | 0.76879 |
| SD (×10⁻⁴) | 0.00050 | 0.00065 | 0.0021 | 0.0039 | 0.0045 | 0.0085 | 0.0087 | 0.00427 | 0.00321 | 0.0072 | 0.00402 |
| CV (=SD/V0) | 0.00284 | 0.00253 | 0.0018 | 0.0013 | 0.0013 | 0.0037 | 0.0044 | 0.00539 | 0.00369 | 0.0051 | 0.00523 |

| Site | Fukuoka | | | | | | | | | | |
|---|---|---|---|---|---|---|---|---|---|---|---|
| Period | 2015/12/4 to 2015/12/20 | | | | | | | | | | |
| No. of days | 8 | | | | | | | | | | |
| SN | PS1202071 Calibrated by Skyradiometer PS1207831 | | | | | | | | | | |

| Wavelength (nm) | 340 | 380 | 400 | 500 | 675 | 870 | 940 | 1020 | 1225 | 1627 | 2200 |
|---|---|---|---|---|---|---|---|---|---|---|---|
| $V_0$ (×10⁻⁴) | 0.18374 | 0.23346 | 1.2332 | 2.9179 | 3.5176 | 2.3021 | 1.9827 | 1.8899 | 0.84113 | 1.2783 | 0.60461 |
| SD (×10⁻⁴) | 0.00028 | 0.00025 | 0.0014 | 0.0025 | 0.0041 | 0.0044 | 0.0106 | 0.0031 | 0.00279 | 0.0027 | 0.00113 |
| CV (=SD/V0) | 0.00155 | 0.00107 | 0.0011 | 0.0008 | 0.0012 | 0.0019 | 0.0053 | 0.0016 | 0.00331 | 0.0021 | 0.00186 |

V0: mean value

SD: standard deviation

CV: coefficient of variation or relative standard deviation (=SD/V0)

Table 4 Statistics of difference between IML method and normal Langley method.

| Wavelength (nm) | 340 | 380 | 400 | 500 | 675 | 870 | 1020 |
|---|---|---|---|---|---|---|---|
| V0 ($\times 10^{-4}$) | 0.17600 | 0.26022 | 1.1840 | 2.9161 | 3.4681 | 2.2863 | 1.2487 |
| BIAS ($\times 10^{-4}$) | −0.00136 | −0.00428 | −0.0042 | 0.0083 | −0.0125 | 0.0017 | 0.0048 |
| RMS ($\times 10^{-4}$) | 0.00325 | 0.00649 | 0.0198 | 0.0225 | 0.0209 | 0.0197 | 0.0309 |
| DFMAX ($\times 10^{-4}$) | 0.00725 | 0.01218 | 0.0368 | 0.0475 | 0.0445 | 0.0489 | 0.0558 |
| DFMIN ($\times 10^{-4}$) | 0.00006 | 0.00059 | 0.0004 | 0.0011 | 0.0028 | 0.0002 | 0.0006 |
| BIAS/V0 | −0.0077 | −0.0164 | −0.0036 | 0.0028 | −0.0036 | 0.0008 | 0.0039 |
| RMS/V0 | 0.0184 | 0.0249 | 0.0167 | 0.0077 | 0.0060 | 0.0086 | 0.0247 |
| DFMAX/V0 | 0.0412 | 0.0468 | 0.0311 | 0.0163 | 0.0128 | 0.0214 | 0.0447 |
| DFMIN/V0 | 0.0003 | 0.0023 | 0.0004 | 0.0004 | 0.0008 | 0.0001 | 0.0005 |
| V0_3RM ($\times 10^{-4}$) | 0.17604 | 0.26037 | 1.1848 | 2.9150 | 3.4678 | 2.2858 | 1.2794 |
| BIAS ($\times 10^{-4}$) | −0.00114 | −0.00389 | −0.0030 | 0.0093 | −0.0124 | 0.0022 | 0.0030 |
| RMS ($\times 10^{-4}$) | 0.00303 | 0.00594 | 0.0178 | 0.0181 | 0.0171 | 0.0163 | 0.1014 |
| DFMAX ($\times 10^{-4}$) | 0.00495 | 0.01065 | 0.0321 | 0.0398 | 0.0352 | 0.0338 | 0.3663 |
| DFMIN ($\times 10^{-4}$) | 0.00022 | 0.00006 | 0.0007 | 0.0032 | 0.0013 | 0.0006 | 0.0004 |
| BIAS/ V0_3RM | −0.0065 | −0.0149 | −0.0025 | 0.0032 | −0.0036 | 0.0010 | 0.0023 |
| RMS/ V0_3RM | 0.0172 | 0.0228 | 0.0150 | 0.0062 | 0.0049 | 0.0071 | 0.0793 |
| DFMAX/ V0_3RM | 0.0281 | 0.0409 | 0.0271 | 0.0137 | 0.0102 | 0.0148 | 0.2863 |
| DFMIN/ V0_3RM | 0.0013 | 0.0002 | 0.0006 | 0.0011 | 0.0004 | 0.0002 | 0.0003 |

V0: mean calibration constant (IML method) during Jan. 2014 to Dec. 2015.

V0_3RM: mean calibration constant (IML method, 3-point running mean) during Jan. 2014 to Dec. 2015.

BIAS: bias (mean of differences between IML and normal Langley methods).

RMS: root mean squares of differences between IML and normal Langley methods.

DFMAX: maximum difference between IML and normal Langley methods.

DFMIN: minimum difference between IML and normal Langley methods.

Table 5 Calibration constants for POM-02 determined by using the calibrated integrating sphere measurement.

| $\lambda_0$ | $\overline{F}_0$ | $\overline{I}_{sph}$ | $\Delta\Omega$ $(\times 10^{-4})$ | $V_{sph}$ $(\times 10^{-10})$ | $V_{sun}$ $(\times 10^{-4})$ | $V_0$ $(\times 10^{-4})$ | $(V_{sun} - V_0)/V_0$ | $I$ |
|---|---|---|---|---|---|---|---|---|
| (nm) | (mW/m²/nm) | (mW/m²/sr/nm) | (sr) | (A) | (A) | (A) | (%) | |
| 340 | 1036.2 | - | 2.3970 | - | - | 0.19884 | - | - |
| 380 | 1210.6 | 24.5 | 2.4370 | 1.9699 | 0.39941 | 0.39280 | 1.68 | PTFE(4.17A(50W)x4) |
| 400 | 1523.3 | 49.6 | 2.4190 | 13.376 | 1.6982 | 1.6434 | 3.34 | PTFE(4.17A(50W)x4) |
| 500 | 1964.6 | 238.1 | 2.4170 | 87.342 | 2.9817 | 2.7703 | 7.63 | PTFE(4.17A(50W)x4) |
| 675 | 1496.5 | 764.1 | 2.4220 | 409.02 | 3.3075 | 3.2850 | 0.69 | PTFE(4.17A(50W)x4) |
| 870 | 958.1 | 1171.1 | 2.4310 | 772.57 | 2.6000 | 2.4708 | 5.23 | PTFE(4.17A(50W)x4) |
| 940 | 822.0 | 1218.8 | 2.4520 | 878.84 | 2.4173 | 2.3364 | 3.46 | PTFE(4.17A(50W)x4) |
| 1020 | 698.1 | 1236.8 | 2.4520 | 682.48 | 1.5710 | 1.5559 | 0.97 | PTFE(4.17A(50W)x4) |
| 1225 | 466.5 | 537.3 | 1.9800 | 204.73 | 0.89767 | 0.88715 | 1.19 | PTFE(3.30A(50W)x4) |
| 1627 | 236.0 | 377.2 | 2.0000 | 459.62 | 1.4378 | 1.4456 | -0.54 | PTFE(3.30A(50W)x4) |
| 2200 | 82.0 | 128.2 | 2.0570 | 237.19 | 0.73756 | 0.72472 | 1.77 | PTFE(3.30A(50W)x4) |

$V_0$: calibration constant by normal Langley method

Table 6    Calibration constant at 940 nm by modified Langley method using the data taken at MLO.

|  |  | Langley | modified Langley | Ratio |
|---|---|---|---|---|
| 2014 | V0 ($\times 10^{-4}$) | 2.3364 | 2.2973 | 0.9833 |
|  | SD ($\times 10^{-4}$) | 0.2183 | 0.1195 |  |
|  | SD/V0 | 0.0934 | 0.0520 |  |
|  | No. of data | 19 | 19 |  |
| 2015 | V0 ($\times 10^{-4}$) | 2.3157 | 2.2954 | 0.9912 |
|  | SD ($\times 10^{-4}$) | 0.2236 | 0.1077 |  |
|  | SD/V0 | 0.0966 | 0.0469 |  |
|  | No. of data | 30 | 20 |  |

The data taken at MLO in 2014 and 2015 were used.

V0: mean value

SD: standard deviation

Ratio = (modified Langley V0)/(Langley V0)

Table 7    Same as Table 6 but using the data taken at Tsukuba, MRI.

|      |                     | Langley | modified Langley | Ratio  |
|------|---------------------|---------|------------------|--------|
| 2014 | V0 ($\times 10^{-4}$) | 2.3343  | 2.3562           | 1.0094 |
|      | SD ($\times 10^{-4}$) | 0.0002  | 0.1429           |        |
|      | SD/V0               | 0.0001  | 0.0598           |        |
|      | No. of data         | 10      | 10               |        |
| 2015 | V0 ($\times 10^{-4}$) | 2.3132  | 2.3090           | 0.9982 |
|      | SD ($\times 10^{-4}$) | 0.0006  | 0.1043           |        |
|      | SD/V0               | 0.0003  | 0.0452           |        |
|      | No. of data         | 17      | 17               |        |

The data taken at MRI, Tsukuba in December 2014 and December 2015 were used.

V0: mean value

SD: standard deviation

Ratio = (modified Langley V0)/(Langley V0)