# Peer review of "Part I: Calibration constant"

_Atmospheric Measurement Techniques, 2017_

## Short Comment (SC1) · 25 Jan 2018

The manuscript reports some very interesting data and application of various methodologies mainly from the the AERONET/SKYNET calibration framework for two instruments but the arguments for some of the results and conclusions are not convincing or not well explained. As in the part II paper there are a significant amount of assumptions and previous results without references that need to be listed. An expanded and re-written manuscript could fix these issues would be most welcome to all interested in atmosphere-based spectral Sun radiometer calibration. A brief summary of the major issues is below. Any manuscript for a global audience needs to conform to some international standards of nomenclature. Unfortunately, the authors use 'accuracy' as a quantitative property for the majority of the paper when 'accuracy' is a qualitative term

(i.e. good, bad, excellent); only in the last parts of the paper is the term 'uncertainty' used but without any explanation of what coverage factor or degrees of freedom. Similarly, the paper quite clearly calls aerosol optical depth (AOD) 'aerosol optical thickness'(AOT) when AOT = m*AOD, and it is only through equation (1) at line 327 that what the authors mean by 'optical thickness' becomes clear. The paper quite clearly demonstrates the 'calibration constant' of the POM-02 is not a constant for the majority of channels (if any) [and likely true for a number of spectral radiometers!]. Instead it may have been useful to define it as the 'coefficient used on a day that represents the signal at the top of the atmosphere at 1'AU and at a representative temperature of X degC'. So why persist in using the term 'constant'? This could have been a key conclusion of the paper rather than implied or assumed. What is a 'normal Langley method'? There are so many variations of 'the Langley method' in the literature that they could be listed on several pages. No reference was provided for the specific method used, and what was more confusing was the application of at least 4 variants of the 'normal' method resulting in Table 1 - and no reference on how the gaseous applications or temperature correction were done, or the reason for the very high standard deviations in an unknown set of MLO calibrations when gaseous and temperature corrections were applied. The non-description of the applied methodologies and the non-explanation of the variances is an example where some references or further detail is required. The temperature coefficients and their application to the raw signals is a key piece of information for other users. However, the section is another example where minimal methodology is presented. There was no experimental setup provided only that it 'was used to measure the temperature dependence of the pyranometer' or the likely uncertainty of the process and the choice of a representative temperature for each sensor. As written, it could almost be assumed that a single value was applied per 'Langley' period rather than individual measurements, and one would have to guess on the representative temperature. It was also disappointing not to see a comparison of the derived coefficients to the sensor manufacturers' specification sheets. The description of the temperature environment in the POM-02 is a very, very useful - though

one could argue that use of the term 'temperature control' was not appropriate. In section 4 (line 180+) the results of the 'normal Langley method' are described in terms of 'errors' but there was no reference only a mean (weighted by an unknown weight or unweighted) hence use of the term 'error' is inappropriate for an unknown parameter of a probability distribution. But an examination of the table suggests these just the (unbiased) standard deviations and therefore only contribute to a single component of the total uncertainty of the 'normal Langley method'. As indicated previously, no indication is given for the increase in this uncertainty component when the sensor signals are corrected (in a manner unknown) when compared to no temperature correction. The lines 232 to 244 describe the likely variation in the 'calibration constant' obtained at MLO over a period of years for the reference POM-02, and summarized on Figure 5 which has a log scale likely because of the range in the Vo values. If the variation is important, then the results should have been scaled to say the 2010 calibration. It would then also be a better lead into the discussion of the interpolation method (and associated uncertainty) that could be required to ensure a required uncertainty (i.e. 2% for high AOD environments for an unknown air mass range - see the WMO (2005) for the working POM-02. The discussion on the reasons for the seasonal variation of the ILM was not convincing, and the lack of opportunity to perform of verification by using calibrating the working instrument with the reference instrument when the seasonal peaks and troughs of the ILM occur was disappointing. Given that the selection of true or apparent solar zenith angle, the airmass type, the rate of change of airmass, and the airmass range used are known to have a seasonal impact on derived 'calibration constants' derived from almost any Langley method variant it was disappointing they could not be examined even for the 500 nm channel of the Tsukuba POM-02. The authors applied a variant if the general method for the calibration of near-infrared channels. It is a pity that it wasn't applied to all wavelengths either using the reference POM-02 AOD or the most stable channel of the Tsukuba POM-02, and hence also test the small variations in wavelength over time. The comparison of the general and ratio (i.e Dobson spectrophotometer) method results to the 'AERONET/SKYNET' methodologies that have not changed since the inception of AERONET and largely based on the hand-held sunphotometer comparison procedures developed at NOAA by Ed Flowers in the 1960s would have been very interesting given the breadth of the excellent JMA data set.

---

## Referee Comment (RC1) · Anonymous Referee #1 · 12 Feb 2018

The instrument constant of sky radiometer (POM-02), Part I: Calibration constant

Akihiro Uchiyama, Tsuneo Matsunaga , Akihiro Yamazaki

Review For Atmospheric Measurement Techniques

General Comments:

This paper overall is a useful contribution to the literature, as it includes discussion of several issues that are often overlooked in sunphotometry, such as the temperature dependence of the detectors. However in order to provide a complete assessment of the uncertainties and issues involved in calibrating sunphotometers, additional information needs to provided and discussed before final publication. One aspect that is lacking is a description of the filters utilized in the POM-02 instruments, such as the bandpass

width of the ion-assisted deposition interference filters for each wavelength, the filter transmittance values and the filter blocking to exclude out-of-bandpass energy. Filter issues such as insufficient blocking can also potentially contribute to calibration uncertainty. Some important information about Langley calibrations done at the Mauna Loa Observatory (MLO) is missing, such as the well-known fact that only morning Langleys should be used for calibration due to unstable conditions in the afternoon as a result of vertical growth of the marine boundary layer to the observatory altitude. References describing the characteristics of the MLO site specifically as related to the Langley calibration method should be added to the manuscript (see Shaw, 1979 JAS; Shaw, 1983 BAMS; Perry et al., 1999 JGR). When discussing the calibration transfer of Vo from a reference instrument to another one in Section 4.2, it is critical to emphasize the importance of the AOD stability during the interval of simultaneous measurements as AOD temporal variability can incur additional uncertainty in Vo transfer. Additional information needs to be included such as how long a time interval was utilized and the time matching criteria used (how many seconds and how many observations matched) for the inter-comparison measurements. Additionally some discussion on how you account for small differences in wavelengths between compared instruments (should use wavelength interpolations) needs to added to the text. Some mention should be made of the fact that near solar noon time intervals are typically the best for calibration transfer since optical airmass (m) changes most slowly at this time and therefore inexact time matching between the instrument measurements is minimized. Another advantage of the use of the solar noon time interval is that if there are differences in filter blocking between instruments then Vo transfers made at the smallest optical airmass are reduced by a factor of 1/m at the larger airmasses. Also, there is larger uncertainty in the computation of optical airmass at large values of optical airmass (see Russell et al., 1993; JGR). I recommend publication of this manuscript in AMT but only after significant revisions that address my general comments, and also after appropriate changes are made to address the specific comments listed below.

Specific comments:

[Figure]

Abstract, Line 15: Please add 'optical properties of' before the word 'aerosols'

Abstract, Line 23: Please mention that the normal Langley method is performed at Mauna Loa Observatory here in the abstract as this is very important information.

Line 111: Remove 'Mt.' as Mauna Loa is never referred to as Mt.

Line 122-123: 'using special equipment' to measure temperature dependence. Please provide much more information on this equipment and on how the measurements are taken with this equipment.

Line 144: Please define 'turret' here, as it is not a commonly used term. I assume it is the rotating filter wheel that holds the individual filters?

Line 156-157: The nomenclature that you have utilized for wavelength regions is poor and not very specific. Note that visible is typically defined as 400 – 700 nm, near-infrared (NIR) as 700 – 1000 nm and shortwave infrared (SWIR) as 1000 - 2500 nm.

Line 181-182: Please give references for the 'normal Langley method' that you refer to here.

Line 187-188: It is well known that afternoon Langley plots at Mauna Loa are much more variable due to marine boundary layer vertical growth. Please note this fact here. It would have been much more robust to use only morning Langleys as the AERONET project does.

Line 194-196: This statement is too general, as some near-infrared channels (such as 870 nm) do not have water vapor absorption.

Line 204-206: You should note that for 380 to 1020 nm the SD/Vo of $\sim$0.2 to 0.5%, very similar to the repeatability values of Vo for AERONET as given in Holben et al. (1998).

Line 206-207: Need to specify how the weighting is done to compute the weighted mean you refer to here.

Line 224-231: Please specify here or in the later section on this topic (section 7) how important it is to account to the vertical profile of water vapor. What is the percentage difference if just an average vertical profile is utilized rather than a specific profile for that date and location?

Line 247-248: What are the channels (give wavelengths) that had annual changes of <1% from 2009 to 2013?

Line 291-294: Please show the monthly mean AOD over the annual cycle and/or add this information to the discussion.

Line 298-300: Please be clear here, are you talking about the difference between the IML and the inter-calibration Vo values?

Line 307-308: Please note that these maximum differences are highly dependent on wavelength.

Line 311-320: Do you have any ideas what may cause the seasonal trends in IML errors? Temperature is accounted for, and AOD is higher in summer when errors are smaller. Possibly optical airmass differences (larger m in winter) in conjunction with filter blocking differences may be bigger factors in winter. Some discussion of possible reasons for the seasonality of IML errors should be added to the text.

Line 395-396: Is this 2% uncertainty based on one standard deviation uncertainty?

Line 400: What is the fixed value that is assumed for the refractive index? Are both real and imaginary parts assumed?

Line 441 – 442: Please give the wavelength ranges here rather than just channel numbers so that the reader does not have to keep referring to the Table when reading the text.

Line 508: Please clarify how you computed the percentage differences in this sentence. Describe more completely what you are talking about here.

Line 525-528: Please note that both AOD and columnar water vapor need to be stable over the full Langley airmass range of measurements. It is very risky to use only one 'stable and fine day' since repeatability cannot be determined and therefore uncertainty cannot be assessed.

Line 531: Please replace 'near-infrared' with 'shortwave infrared'.

Line 680-681: Please state the wavelength of this channel that has the maximum error.
* * *

---

## Referee Comment (RC2) · Anonymous Referee #2 · 1 May 2018

The paper describes in-field and laboratory calibrations of POM-02 sky radiometers used by SKYNET aerosol Network. The first method includes Langley-plot "zero airmass" intercept determination at high altitude Mauna Loa observatory for the POM-02 (Calibration reference) instrument with following calibration transfer to other instruments on-site (e.g., at Tsukuba site). The second on-site calibration method includes Improved Langley calibration (IML), without using reference instrument. Temperature effect on the calibration constant is shown to be important in the UV (340nm and 380nm) and shortwave infrared (2200nm) spectral channels. The temperature sensitivity varies for different instruments. The temperature effect on visible and NIR channels is generally small (< 0.5% for a typical temperature range).

The paper is of general interest for ground-based aerosol measurement community

and could be published after major revision.

General comments: The main manuscript should be clarified focusing on main conclusions, while supporting material (technical details, tables and plots) could be moved to the supplement. Clarify calibration adjustment to account for changing sun-Earth distance. English should be improved. References need to be updated. Figure quality needs improvements

Specific comments: Describe how spectral response functions were measured. Replace "near-infrared" with "shortwave infrared": for > 1-micron channels Suggest replacing "SVA" with commonly used Field of View (FOV)

Improved Langley method (IML) should be clearly explained – see comments L359-379

L21: indicate temperature climatology in Tsukuba L23: Is this accuracy at Tsukuba or Mauna Loa? L25: quantify V0 uncertainty in UV-VIS-NIR and degradation (V0 time drift?) L26: Clarify that this is accuracy of calibration transfer only during best stable atmospheric conditions. Indicate time intervals for calibration transfer L33: change to short infrared L35: this sentence does not belong to the abstract L37: Quantify accuracy for each channel.

L59: Column average effective aerosol characteristics . . . L68: add references L71: SVA is usually called Field of View (FOV) L74: Provide instrumental reference. L135: which temperature sensor is used to start the heater? L136: ". . . the instrument is heated. . ." – to what temperature? When does the heater stop ?

L139: use "shortwave infrared" L142: inside temperature[s]? L146, Fig 1: Explain why if the temperature control setting was 20C , the inside temperature was 30C when the ambient temperature was 20C?

L153: . . . wavelengths shorter than 1020nm . . .

L 157 "near-infrared region" – common name is "shortwave -infrared region"

L161: "change in the temperature less than 1.5%" -> "change in the instrument response less than 1.5%"?

L188, Fig.4: Specify units in Y axis , e.g. counts per second? – this is usually a large number: explain scaling.

L197-248, Table 1: Explain units for calibration constant (V0) ? Table 1: Explain if correction for changing Sun-Earth distance was applied to daily V0s?.

L204 The [standard] error L210 "is large . . ." – quantify L222" without consideration of the temperature . . ." –for MLO conditions only L233"was replaced" -> were replaced L240: What are reasons for such large V0 changes ? L244: It would be useful to show monthly V0 values (corrected for sun-Earth distance) in fig.5.

L250: "Accuracy of [V0 calibration] transfer by direct solar measurement"

267 5. Improved Langley method - Add paragraph describing IML here

L289 IML value[s] L295: delete "by" L330 layer -> atmospheric column optical thickness

322 5.2 Review of Improved Langley method – move this section up after L267

325 The solar direct irradiance at the surface [normal to the solar beam] 330 zenith angle, and [tau] is the layer optical thickness. – total atmospheric optical thickness (Rayleigh plus aerosol plus gases) 331 The single scattering by aerosol in the almucantar - replace "by aerosol plus molecular (Rayleigh) scattering in the almucantar "

L340 direct solar [voltage] measurement L341: Equation (3) neglects forward scattered radiation into the FOV L341-345: Equations (3) and (4) should include Earth-sun distance (see eq. (19) )

L353: Explain how tau or tau_scat can be obtained independently from the V0 ? L359: Explain how tau is obtained ? L360: Explain how tau_scat is obtained? L367: SVA -> FOV (common name) L368: ".. is the [radiometer output (voltage) due to ] direct solar irradiance [at the surface]

L375: "Once the single scattering component is retrieved, m*tau and m*tau_scat are estimated" - Solving radiation transfer equation is only possible if tau, Phase function and tau_scat are known. Explain how tau, P(scat) and tau_scat are obtained?

What are introduced uncertainties due to assumptions about unknown aerosol refractive index, size distribution, modeling of aureole forward scattered radiation?

379 "Once m*tau is obtained," – explain how tau is obtained before knowing calibration constant V0? Is another co-located radiometer used to derive tau?

L380-381 do not use capital for single scattering albedo: W0 L384 "W0 is frequently greater than 1." – are these unphysical retrievals used for calibration? L387 Figs. -> Fig.9 L389 "V0 values with errors less than 0.01" – Is this error in ln(V0) ? L393, Fig9(c): In this plot was V0 corrected for the changing sun-Earth distance? L398 " . . . are systematically overestimated". – please, clarify this statement

ÂňÂň L414 " [and spectral response function of the ] radiometer are necessary"

L438: Table 4: Provide units for Vsun and V0

L444-455: Fig 10: Compare with more recent sources of high spectral resolution extra-terrestrial solar irradiance, e.g. https://www.cfa.harvard.edu/atmosphere/publications/Chance-Kurucz-solar2010-JQSRT.pdf

L466: which takes [into] account . . . L488-490: Use tau_aer in Eq (19) and 490 L492. ".. is interpolated from the optical thicknesses at 870 and 1020 nm" – explain interpolation method, e.g. linear, power law?

L496: explain how R is calculated? L497-499: explain how coefficients a and b were calculated? L504-505: explain units for calibration coefficients? L548: use tau_aer L640: "seasonal variation of 1 to 3%." – Correcting for sun-Earth distance ?

Technical comments:

L559-560: Equations (24) and (25) can be combined. L585: "... is an alternative to the Langley method." – extension of Langley method? L629: " The annual variation of the calibration constants..." – The long-term changes

Fig 1 caption: "inside temperature[s]" Fig.4. Check the Y units: be counts per second? What is the scaling factor? Fig.5. Show monthly V0 values to check V0 seasonal dependence Fig. 8 Too small axis labels. Suggest scaling Y axis for clarity

---

## Author Comment (AC1) · 18 Jun 2018

Reply to comments

We would like to thank you for reading our manuscript and commenting on it.

The comments are copied and shown below in italic.

*Comment.*

*The manuscript reports some very interesting data and application of various methodologies mainly from the the AERONET/SKYNET calibration framework for two instruments but the arguments for some of the results and conclusions are not convincing or not well explained. As in the part II paper there are a significant amount of assumptions and previous results without references that need to be listed. An expanded and re-written manuscript could fix these issues would be most welcome to all interested in atmosphere-based spectral Sun radiometer calibration.*

*A brief summary of the major issues is below. Any manuscript for a global audience needs to conform to some international standards of nomenclature. Unfortunately, the authors use 'accuracy' as a quantitative property for the majority of the paper when 'accuracy' is a qualitative term (i.e. good, bad, excellent); only in the last parts of the paper is the term 'uncertainty' used but without any explanation of what coverage factor or degrees of freedom.*

==>

Reply

We rewrote carefully and added explanations.

*Similarly, the paper quite clearly calls aerosol optical depth (AOD) 'aerosol optical thickness'( AOT) when AOT = m\*AOD, and it is only through equation (1) at line 327 that what the authors mean by 'optical thickness' becomes clear.*

==>

Reply

We replaced "optical thickness" with "optical depth".

*The paper quite clearly demonstrates the 'calibration constant' of the POM-02 is not a constant for the majority of channels (if any) [and likely true for a number of spectral radiometers!]. Instead it may have been useful to define it as the 'coefficient used on a day that represents the signal at the top of the atmosphere at 1'AU and at a*

*representative temperature of X degC'. So why persist in using the term 'constant'? This could have been a key conclusion of the paper rather than implied or assumed.*

==>

Reply

In this paper, calibration constant V0 is the output of the radiometer to the extra-terrestrial solar irradiance at the mean earth-sun distance (1 AU) at reference temperature. However, when comparing V0 by the IML method with V0 based on the Langley method, the temperature correction is not performed.

We conventionally use the word "calibration constant". But, the sensor output depends on the temperature and it varies with the aging of the radiometer. Therefore, V0 varies with time.

The solar constant also changes, but it is called the solar constant.

*What is a 'normal Langley method'? There are so many variations of 'the Langley method' in the literature that they could be listed on several pages. No reference was provided for the specific method used, and what was more confusing was the application of at least 4 variants of the 'normal' method resulting in Table 1 - and no reference on how the gaseous applications or temperature correction were done, or the reason for the very high standard deviations in an unknown set of MLO calibrations when gaseous and temperature corrections were applied. The non-description of the applied methodologies and the non-explanation of the variances is an example where some references or further detail is required.*

==>

Reply

We wrote a brief explanation on the Langley method and described the method we did.

*The temperature coefficients and their application to the raw signals is a key piece of information for other users. However, the section is another example where minimal methodology is presented. There was no experimental setup provided only that it ' was used to measure the temperature dependence of the pyranometer' or the likely uncertainty of the process and the choice of a representative temperature for each sensor. As written, it could almost be assumed that a single value was applied per 'Langley' period rather than individual measurements, and one would have to guess on the representative temperature. It was also disappointing not to see a comparison of the*

*derived coefficients to the sensor manufacturers' specification sheets. The description of the temperature environment in the POM-02 is a very, very useful – though one could argue that use of the term 'temperature control' was not appropriate.*

==>

Reply

We added about the measurement procedure of temperature characteristics.

The temperature correction was performed for each measurement data.

The temperature dependence of the sensor output is approximated by the following equation.

$$V(T)/V(T=Tr) = 1.0 + C_1(T-Tr) + C_2(T-Tr)^2$$

where $V(T)$ is sensor output at temperature $T$, $V(T=Tr)$ is sensor output at reference temperature $Tr$, $Tr$ is reference temperature, coefficients $C_1$ and $C_2$ are determined by the least squares method.

Therefore, measured $V(T)$ is corrected by the following equation.

$$V(T=Tr) = V(T)/(1.0 + C_1(T-Tr) + C_2(T-Tr)^2)$$

Instruments are designed to operate the heater when the inside temperature is less than 20 or 30 ºC. For colder regions such as polar regions, the setting temperature is 20 ºC, and in other regions, the setting temperature is 30 ºC. When the temperature near the rotating filter wheel inside the instrument is below its threshold temperature (20 or 30 ºC), the instrument is heated. When the temperature exceeds the threshold, heating is stopped. However, there is no cooling mechanism for when the temperature inside the instrument is higher than its threshold temperature for optimum operation. The temperature control setting of POM-02 (Calibration Reference) is 20 ºC, and that of POM-02 (Tsukuba) is 30 ºC.

According to the specification sheet, the temperature dependence of the detector sensitivity is almost zero (cannot be read) at wavelengths from 300 nm to 950 nm. At a wavelength of 1020 nm, it is about 0.2%/deg. At wavelengths of 1225 nm, 1627 nm and 2200 nm, they are almost zero, $-0.05$, 0.02%/deg, respectively. The temperature dependences of sensor output shown in Figs. 2 and 3 are the characteristic of the entire instrument. Some channels exhibit greater temperature dependence than the temperature dependence of the detector.

*In section 4 (line 180+) the results of the 'normal Langley method' are described in terms of 'errors' but there was no reference only a mean (weighted by an unknown weight or unweighted) hence use of the term 'error' is inappropriate for an unknown parameter of a probability distribution. But an examination of the table suggests these just the (unbiased) standard deviations and therefore only contribute to a single component of the total uncertainty of the 'normal Langley method'.*

*As indicated previously, no indication is given for the increase in this uncertainty component when the sensor signals are corrected (in a manner unknown) when compared to no temperature correction.*

==>

Reply

The term "error" is not an appropriate word. According to comment from other reviewer, it was rewritten as "relative standard deviation or coefficient of variation". In any case, as we mentioned in the text, it is SD / V0; SD:standard deviation and V0 is mean value.

*The lines 232 to 244 describe the likely variation in the 'calibration constant' obtained at MLO over a period of years for the reference POM-02, and summarized on Figure 5 which has a log scale likely because of the range in the Vo values. If the variation is important, then the results should have been scaled to say the 2010 calibration. It would then also be a better lead into the discussion of the interpolation method (and associated uncertainty) that could be required to ensure a required uncertainty (i.e. 2% for high AOD environments for an unknown air mass range - see the WMO (2005) for the working POM-02.*

==>

Reply

In Fig. 5, we only want to show that V0 changes with time and its rate of change depends on the wavelength. We would like to keep this figure as it is.

*The discussion on the reasons for the seasonal variation of the ILM was not convincing, and the lack of opportunity to perform of verification by using calibrating the working instrument with the reference instrument when the seasonal peaks and troughs of the ILM occur was disappointing. Given that the selection of true or apparent solar zenith*

*angle, the airmass type, the rate of change of airmass, and the airmass range used are known to have a seasonal impact on derived 'calibration constants' derived from almost any Langley method variant it was disappointing they could not be examined even for the 500 nm channel of the Tsukuba POM-02.*

==>

Reply

Until now, no one has pointed out that the V0 determined by the IML method changes seasonally. First of all, we want POM-02 users to know this fact. As you say, many factors are changing seasonally. It is very difficult to clarify which seasonal changes are related to the seasonal change of V0 determined by IML method.

Let the difference between V0 determined by the IML method and V0 interpolated from V0 determined by inter-comparison with the reference POM-02 be ΔV0.

Since ΔV0 is correlated with the optical thickness and the fitting coefficient of the IML method, we believe that V0 determined by the IML method is related to optical thickness and refractive index.

The Improvement of the IML method is a future task.

*The authors applied a variant if the general method for the calibration of near-infrared channels. It is a pity that it wasn't applied to all wavelengths either using the reference POM-02 AOD or the most stable channel of the Tsukuba POM-02, and hence also test the small variations in wavelength over time. The comparison of the general and ratio (i.e Dobson spectrophotometer) method results to the 'AERONET/SKYNET' methodologies that have not changed since the inception of AERONET and largely based on the hand-held sunphotometer comparison procedures developed at NOAA by Ed Flowers in the 1960s would have been very interesting given the breadth of the excellent JMA data set.*

==>

Reply

We applied this method to all channels and obtained similar results. However, we are interested in the 1225, 1627, and 2200 nm channels which are not calibrated by SKYNET. Therefore, we showed results only for them.

The method we tried in this study is not used in SKYNET. We have a plan to use SKYNET data to verify the aerosol optical thickness retrieved from the satellite data. The aerosol optical thickness in the shortwave-infrared region cannot be estimated in

SKYNET. Therefore, we tried this method by ourselves.

The data used here is the data taken by JMA research branch (JMA/MRI). It is not the data taken by JMA routine observation branch. The JMA routine observation branch recently started observing with POM-02, but they do not have the technique to calibrate POM-02 by themselves. They participated in the 4th WMO Filter Radiometer Comparison (Kazadzis et al. 2018). Although we are not in the author of this paper, the calibration constant of POM-02 used by them was transferred from the POM-02 (Calibration Reference) in this study by the method shown in this paper. In this inter-comparison campaign, aerosol optical thicknesses at the wavelength of 500 nm and 875 nm were compared. The results of comparison showed that JMA's POM-02 achieved WMO criterion (WMO, 2005).   We believe that the method shown in this study is adequate.

---

## Author Comment (AC2) · 18 Jun 2018

Reply to comments

We would like to thank you for reading our manuscript and commenting on it.
The comments are copied and shown below in italic.

*Comment.*
*Anonymous Referee #1*

*The instrument constant of sky radiometer (POM-02), Part I: Calibration constant*
*Akihiro Uchiyama, Tsuneo Matsunaga , Akihiro Yamazaki*
*Review For Atmospheric Measurement Techniques*

*General Comments:*
*This paper overall is a useful contribution to the literature, as it includes discussion of several issues that are often overlooked in sunphotometry, such as the temperature dependence of the detectors. However in order to provide a complete assessment of the uncertainties and issues involved in calibrating sunphotometers, additional information needs to provided and discussed before final publication.*
*One aspect that is lacking is a description of the filters utilized in the POM-02 instruments, such as the bandpass width of the ion-assisted deposition interference filters for each wavelength, the filter transmittance values and the filter blocking to exclude out-of-bandpass energy. Filter issues such as insufficient blocking can also potentially contribute to calibration uncertainty.*
==>
Reply
We add the new Table 1 to show the nominal specification of filter, and insert the following sentence after line 107.
"In Table 1, the nominal specification of filters is shown. JMA / MRI does not use thee 315 nm channel because the transmittance of the lens was low at this wavelength region. Instead, JMA / MRI added a 1225 nm channel."

Table ### Nominal filter specification

| Channel No. | Wavelength (nm) | FWHM(nm) | Max. Transmittance | Blocking | Blocking wavelength | Detector |
|---|---|---|---|---|---|---|
| – | 315($\pm$0.6nm)* | 3.0($\pm$0.6nm) | >30% | $1.0\times10^{-5}$ | 200 – 1200 nm | Si photodiode |
| 1 | 340($\pm$0.6nm) | 3.0($\pm$0.6nm) | >30% | $1.0\times10^{-5}$ | 200 – 1200 nm | Si photodiode |
| 2 | 380($\pm$0.6nm) | 3.0($\pm$0.6nm) | >30% | $1.0\times10^{-5}$ | 200 – 1200 nm | Si photodiode |
| 3 | 400($\pm$0.6nm) | 10.0($\pm$2.0nm) | >30% | $1.0\times10^{-5}$ | 200 – 1200 nm | Si photodiode |
| 4 | 500($\pm$2.0nm) | 10.0($\pm$2.0nm) | >30% | $1.0\times10^{-5}$ | 200 – 1200 nm | Si photodiode |
| 5 | 675($\pm$2.0nm) | 10.0($\pm$2.0nm) | >30% | $1.0\times10^{-5}$ | 200 – 1200 nm | Si photodiode |
| 6 | 870($\pm$2.0nm) | 10.0($\pm$2.0nm) | >30% | $1.0\times10^{-5}$ | 200 – 1200 nm | Si photodiode |
| 7 | 940($\pm$2.0nm) | 10.0($\pm$2.0nm) | >30% | $1.0\times10^{-5}$ | 200 – 1200 nm | Si photodiode |
| 8 | 1020($\pm$2.0nm) | 10.0($\pm$2.0nm) | >30% | $1.0\times10^{-5}$ | 200 – 3000 nm | Si photodiode |
| 9 | 1225($\pm$2.0nm)** | 20.0($\pm$2.0nm) | >30% | $1.0\times10^{-5}$ | 600 – 3000 nm | InGaAs photodiode |
| 10 | 1627($\pm$2.0nm) | 20.0($\pm$2.0nm) | >30% | $1.0\times10^{-5}$ | 600 – 3000 nm | InGaAs photodiode |
| 11 | 2200($\pm$2.0nm) | 20.0($\pm$2.0nm) | >30% | $1.0\times10^{-5}$ | 600 – 3000 nm | InGaAs photodiode |

FWHM : Full Width at Half Maximum

* : 315 nm channel is not used by JMA/MRI.

** : 1225 nm channel is used   by JMA/ MRI.

*Some important information about Langley calibrations done at the Mauna Loa Observatory (MLO) is missing, such as the well-known fact that only morning Langleys should be used for calibration due to unstable conditions in the afternoon as a result of vertical growth of the marine boundary layer to the observatory altitude. References describing the characteristics of the MLO site specifically as related to the Langley calibration method should be added to the manuscript (see Shaw, 1979 JAS; Shaw, 1983 BAMS; Perry et al., 1999 JGR).*

==>

Reply

We added the following sentences in new section 4.2 Normal Langley method.

"Measurements for calibration by the Langley method are recommended to be conducted at a high mountain observatory. MLO is one of the most suitable places to make measurements for calibration by the Langley method. Though the air at MLO is exceedingly transparent, it is affected in late morning and afternoon hours by marine aerosol that reaches the observatory as the marine inversion boundary layer breakdown under solar heating. Typically, by late morning the downslope winds switch to upslope winds, which bring moisture and aerosol-rich marine boundary layer air up the mountainside, resulting in an abundance of orographic clouds at the observatory (Show 1983, Perry et al. 1999). Therefore, using data taken in the morning is recommended and used (Show 1982, Dutton et al. 1994, Holben et al 1998).

In AERONET, the variability of the determined calibration coefficient as measured by the relative standard deviation or the coefficient of variation (RSD or CV, standard deviation/mean) is ~0.25−0.50% for the visible and near-infrared wavelength, ~0.5−2% for the ultraviolet and ~1−3% the for water vapor channel (Holben et al 1998).

In this study, though using data taken in the morning is recommended, both morning and afternoon data were used for the Langley plot. The observation period for calibration by Langley method is short, about 1 month, so we want to use all the data effectively. Furthermore, the quality of the Langley plot can be checked by an analysis of residuals; for acceptable data, no trend or systematic pattern is visible when the residuals versus airmass are plotted. Of course, the residuals were carefully checked and most results of the afternoons data were not adopted."

We also added the following sentences to the explanation of Fig.4.
"In these examples, the data in the afternoon is almost on the regression line in the morning. On such a day, the Langley plot was also applied to the afternoon data."

*When discussing the calibration transfer of Vo from a reference instrument to another one in Section 4.2, it is critical to emphasize the importance of the AOD stability during the interval of simultaneous measurements as AOD temporal variability can incur additional uncertainty in Vo transfer. Additional information needs to be included such as how long a time interval was utilized and the time matching criteria used (how many seconds and how many observations matched) for the inter-comparison measurements. Additionally, some discussion on how you account for small differences in wavelengths between compared instruments (should use wavelength interpolations) needs to added to the text. Some mention should be made of the fact that near solar noon time intervals are typically the best for calibration transfer since optical airmass (m) changes most slowly at this time and therefore inexact time matching between the instrument measurements is minimized. Another advantage of the use of the solar noon time interval is that if there are differences in filter blocking between instruments then Vo transfers made at the smallest optical airmass are reduced by a factor of 1/m at the larger airmasses. Also, there is larger uncertainty in the computation of optical airmass at large values of optical airmass (see Russell et al., 1993; JGR).*
==>
Reply
   Though the temporal AOD stability is one of the important factors, we believe that

simultaneity of data acquisition is important. We added sentences about data acquisition and the comparison method after line 253.

"The measurements for the comparison were made every minute using the same data acquisition system. It takes about 10 seconds to measure 11 channels each time. Measurement by all POM-02 is done at the same timing. Calibration of time is carried out every hour using NTP (Network Time Protocol) Server. For data comparison, only airmass data less than 2.5 was used on clear days. The comparisons were made on the assumption that the filter response function of POM-02 are same. When there is a difference in the filter, the relationship between the outputs of both becomes not linear. When it is greatly deviated from the linear relationship, the characteristics of either filter has changed, and it is necessary to replace the filter."

*I recommend publication of this manuscript in AMT but only after significant revisions that address my general comments, and also after appropriate changes are made to address the specific comments listed below.*

*Specific comments:*
*Abstract, Line 15: Please add 'optical properties of' before the word 'aerosols'*
==>
Reply
  We add 'optical properties of' before the word 'aerosols' .

*Abstract, Line 23: Please mention that the normal Langley method is performed at Mauna Loa Observatory here in the abstract as this is very important information.*
==>
Reply
We changed the sentence as follows.
"The coefficient of variation (CV) of V0 from the normal Langley method based on the data measured at NOAA Mauna Loa Observatory is between 0.2 and 1.3%, except in the 940 nm channel."

*Line 111: Remove 'Mt.' as Mauna Loa is never referred to as Mt.*
==>

Reply

We remove "Mt."

*Line 122-123: 'using special equipment' to measure temperature dependence. Please provide much more information on this equipment and on how the measurements are taken with this equipment.*

= = >

Reply

The word "special" was not appropriate.

As written in the manuscript, this equipment is used originally to measure the temperature dependence of the pyranometer. This equipment is managed and maintained by a branch of the JMA Observation Department, which is one of the departments conducting routine observations.

We delete "special" and added the following sentences to explain measurements for temperature characteristics.

   "The main components of this equipment are a temperature controlled chamber, light source, and stabilized power supply.

   Measurements for investigating the temperature characteristics of POM-02 were made as follows.

   In order to stabilize the equipment, the power supply of the equipment was put on the day before the measurement date. On the measurement day, first turn on the light source. Then, temperature setting is performed every 90 minutes, and temperature and output from POM-02 are recorded continuously. Temperature setting was performed in the order of 40, 20, 0, −20, 0, 20, 40, 20 °C. It took about 30 minutes for temperature rise and about 40 minutes for temperature decrease until the temperature and the output of POM-02 became stable. Temperature characteristics were investigated using data between 70 and 90 minutes.

   In order to check the stability of the equipment, the staff of JMA recorded the output of the pyranometer CMP-22 (Kipp & Zonen, Netherland) continuously for 11 hours at a temperature setting of 20 °C. As a result, the variation of the mean values of the output per hour was ± 0.05% or less.

   The temperature correction was performed for each measurement data. The temperature dependence of the sensor output was approximated by the following equation.

$$V(T)/V(T=Tr) = 1.0 + C_1(T-Tr) + C_2(T-Tr)^2 \qquad\qquad (1)$$

where $V(T)$ is sensor output at temperature $T$, $V(T=Tr)$ is sensor output at reference temperature $Tr$, $Tr$ is reference temperature, coefficients $C_1$ and $C_2$ were determined by the least squares method. Therefore, measured $V(T)$ is corrected by the following equation.

$$V(T=Tr) = V(T)/(1.0 + C_1(T-Tr) + C_2(T-Tr)^2) \qquad\qquad (2)\text{''}$$

*Line 144: Please define 'turret' here, as it is not a commonly used term. I assume it is the rotating filter wheel that holds the individual filters?*

==>

Reply

  We replace "filter turret" with "rotating filter wheel that holds the individual filters".

*Line 156-157: The nomenclature that you have utilized for wavelength regions is poor and not very specific. Note that visible is typically defined as 400 – 700 nm, nearinfrared (NIR) as 700 – 1000 nm and shortwave infrared (SWIR) as 1000 - 2500 nm.*

==>

Reply

  We changed the nomenclature according to your advice.

*Line 181-182: Please give references for the 'normal Langley method' that you refer to here.*

==>

Reply

I explained the Langley method we did in new section 4.1 and 4.2.

The term "normal Langley method" is used in Reagan et al (1986) and Kazadzis et al (2018). In the former, it is explained that the same airmass $m_R$ is assumed for all attenuators, where $m_R$ is airmass for molecular scattering. In the latter, there is no explanation.

*Line 187-188: It is well known that afternoon Langley plots at Mauna Loa are much more variable due to marine boundary layer vertical growth. Please note this fact here. It would have been much more robust to use only morning Langleys as the AERONET project does.*

==>

Reply

See above. We have already explained.

*Line 194-196: This statement is too general, as some near-infrared channels (such as 870 nm) do not have water vapor absorption.*

==>

Reply

We are writing about 1225, 1627 and 2200 nm channels here.

We replaced "near-infrared" with "shortwave-infrared".

*Line 204-206: You should note that for 380 to 1020 nm the SD/Vo of ~0.2 to 0.5%, very similar to the repeatability values of Vo for AERONET as given in Holben et al. (1998).*

==>

Reply

We add the following sentence after Line 209.

"In AERONET, the similar results were obtained (Holben et al. 1998)."

*Line 206-207: Need to specify how the weighting is done to compute the weighted mean you refer to here.*

==>

Reply

We recalculate weighted mean and standard deviation.

We attached an explanation of the weight to the appendix.

*Line 224-231: Please specify here or in the later section on this topic (section 7) how important it is to account to the vertical profile of water vapor. What is the percentage*

*difference if just an average vertical profile is utilized rather than a specific profile for that date and location?*

==>

Reply

In Uchiyama et al (2014), the transmittance of 940nm channel is calculated using the vertical profile of water vapor. In this section, we do not use the modified Langley method. The fluctuation of water vapor is large, and using the average vertical profile, the transmittance cannot be calculated accurately and the Langley plot cannot be done. The modified Langley method requires airmass for water vapor. At that time, it may be useful to use the average vertical distribution

*Line 247-248: What are the channels (give wavelengths) that had annual changes of <1% from 2009 to 2013?*

==>

Reply

Here, we wrote about the channel of shortwave-infrared channels (1225, 1627, 2200 nm). We replace "near-infrared region" with "shortwave-infrared channels (1225, 1627, 220 nm)" in line 245, and add "in the shortwave-infrared channels" in line 247.

*Line 291-294: Please show the monthly mean AOD over the annual cycle and/or add this information to the discussion.*

==>

Reply

Roughly speaking, the optical thickness is thick in summer and thin in winter at Tsukuba. However, I do not know if the statistics on the day when Improved Langley method is applied are the same.

We rewrote Fig. 9. In the new Fig.9, since the error of $V_0$ and the optical thickness were found to be correlated, we do not show the annual cycle of optical thickness here.

In the old Fig.9, $V_0$ determined by the IML method was directly compared with the optical thickness. In this comparison, the trend of $V_0$ is not excluded, so the relationship between $V_0$ determined by the IML method and the optical thickness was not clear. For

this reason, we investigated the relationship between $\Delta V_0$ and the optical thickness of

the day when the IML method was applied, where $\Delta V_0$ is the difference between $V_0$

determined by the IML method and $V_0$ interpolated from $V_0$ determined by the normal

Langley method. As a result, it was found that there was a correlation between the two. This result is consistent with the large amplitude of the seasonal change at short wavelengths (the shorter the wavelength, the optically thicker). Therefore, the optically thicker the accuracy of the multiple scattering estimation is poor. And the accuracy of the IML method may be poor. However, since the differences also depend on single scattering albedo, we cannot explain all of the errors with optical thickness.

We do not show the annual cycle of optical thickness, but we rewrote Fig.9 and added the above contents.

*Line 298-300: Please be clear here, are you talking about the difference between the IML and the inter-calibration Vo values?*
==>
Reply
Yes, we are.
We rewrote line 299 and 300.

"The calibration constant $(V_0)$ determined by the IML method changes by up to 6%.

Even if the effect of the temperature change is subtracted from the seasonal variation,

there is a difference of about 4% between the $V_0$ determined by IML method and $V_0$

interpolated from $V_0$ determined by inter-comparison with the POM-02 (Calibration

Reference)."

*Line 307-308: Please note that these maximum differences are highly dependent on wavelength.*
==>
Reply
Yes, they depend on the wavelength.

Since the differences depend on the optical thickness and usually the shorter the wavelength, the thicker the optical thickness, so the shorter the wavelength, the larger the amplitude of the seasonal change of V0 by the IML method. Therefore Max. Difference can be large.

We add the above sentences to the text.

*Line 311-320: Do you have any ideas what may cause the seasonal trends in IML errors? Temperature is accounted for, and AOD is higher in summer when errors are smaller. Possibly optical airmass differences (larger m in winter) in conjunction with filter blocking differences may be bigger factors in winter. Some discussion of possible reasons for the seasonality of IML errors should be added to the text.*

==>

Reply

We said that the value of V0 by IML method is small in the summer and large in the winter. However, we did not say that the error (difference) is small in the summer and large in the winter (see Fig. 8).

We do not know exactly the cause, but We want to show facts and draw attention to users of IML method.

The optical thickness changes seasonally, and seasonal change of V0 of IML method seems to be related to optical thickness. The V0 of the IML method also depends on W0, and simply the optical thickness is not the cause of the error.

Since W0 is a parameter related to single scattering albedo, I think there is a possibility that the seasonal variation of V0 by IML method may also be related to the seasonal variation of the refractive index. In the current processing, since the refractive index is fixed, I think that it is necessary to try a method to determine V0 while changing the refractive index.

We added this content to the explanation of the new Fig. 9.

*Line 395-396: Is this 2% uncertainty based on one standard deviation uncertainty?*

==>

Reply

We delete this sentence. We only calculated the statistics of difference between IML V0 and the inter-calibrated V0.

We replace Fig.9 with new Fig. 9. Therefore, we rewrote the three paragraphs from line

387 to line 403 in section 5.2 and moved them after line 321. And, we moved the rest of section 5.2 to the beginning of section 5.

*Line 400: What is the fixed value that is assumed for the refractive index? Are both real and imaginary parts assumed?*

==>

Reply

We use (1.5, -0.001) for all wavelengths as initial value of refractive index when using the Skyrad package. This value was used here. Since V0 determined by the IML method depended on W0, this value of refractive index may not be appropriate. We will consider the method of determining V0 while changing the refractive index in the future.

We added the above contents.

*Line 441 – 442: Please give the wavelength ranges here rather than just channel numbers so that the reader does not have to keep referring to the Table when reading the text.*

==>

Reply

We rewrote lines 441 and 442 as follows.

"At POM-02 (Calibration Reference), the relative difference was 0.7 to 7.6% in channels 2 to 8 (380 to 1020 nm), and 0.5 to 1.8% in channels 9 to 11 (1225, 1627, and 2200 nm). The integrating sphere used in channels 2 to 8 is different from that in channels 9 to 11."

*Line 508: Please clarify how you computed the percentage differences in this sentence. Describe more completely what you are talking about here.*

==>

Reply

We rewrote as follows.

"Though the difference of calibration coefficient between the Langley method with consideration of gas absorption and the modified Langley method is 1.7% ($2.2973 \times 10^{-4}/2.3364 \times 10^{-4} - 1 = -0.0167$) in 2014 and 0.9% ($2.2954 \times 10^{-4}/2.3157 \times 10^{-4} - 1 = -0.0087$) in 2015, these calibration coefficients are very similar."

*Line 525-528: Please note that both AOD and columnar water vapor need to be stable over the full Langley airmass range of measurements. It is very risky to use only one 'stable and fine day' since repeatability cannot be determined and therefore uncertainty cannot be assessed.*

==>

Reply

Assuming that V0 at 875 nm and 1020 nm are known, AOD at 940 nm is interpolated from AOD at 875 nm and 1020 nm. We only assume that pwv is constant.

By checking the residuals of the regression line, we can check whether the calibration constant is determined accurately. The 940 nm channels at many observation sites in SKYNET have not been calibrated and are not used. The application of the modified Langley method to the on-site observation data is the next best solution.

We add the following sentences after line 528.

"The quality of the Langley plot can be checked by an analysis of residuals; for acceptable data, no trend or systematic pattern is visible when the residuals versus airmass are plotted. The 940 nm channels at many observation sites have not been calibrated and are not used. The application of the modified Langley method to the on-site observation data is the next best solution".

*Line 531: Please replace 'near-infrared' with 'shortwave infrared'.*

==>

Reply

We replaced 'near-infrared' with 'shortwave infrared'.

*Line 680-681: Please state the wavelength of this channel that has the maximum error.*

==>

Reply

*We add channel no. and wavelength.*

Line 389-403。

We redrew Fig. 9 and rewritten the text as follows.

For the 500 nm channel, Figs. 9 shows a scatter plot of $\Delta V_0$ and the optical depth at

500 nm, a scatter plot of $\Delta V_0$ and $W_0$, and a time series of $\Delta V_0$ from January 2014 to December 2015, where $\Delta V_0$ is the difference between $V_0$ determined by the IML method and $V_0$ interpolated from $V_0$ determined by inter-comparison with the POM-02 (Calibration Reference). In this case, the $V_0$ values by IML method with errors less than 0.01 were chosen, where error is root mean square difference between measurement value and fitting line. As shown in Fig. 8, Fig. 9 (c) shows that $\Delta V_0$ changes seasonally.

 Figure 9 (a) shows that there is a negative correlation between $\Delta V_0$ and the optical depth. This result is consistent with the large amplitude of the seasonal change at short wavelengths. Since usually the shorter the wavelength, the thicker the optical depth, so the shorter the wavelength, the larger the amplitude of the seasonal change of $V_0$ by the IML method.

 In Tsukuba, the aerosol optical depth is thick in the summer and thin in the winter. Therefore, the seasonal change of $V_0$ by the IML method seems to be related to optical thickness. However, Fig. 9 (b) also shows that $\Delta V_0$ and $W_0$ are negatively correlated, and that even if the correct $W_0$ is determined, the $\Delta V_0$ are scattered with a width of about $1.0 \times 10^{-5}$. Since $W_0$ is a parameter related to single scattering albedo or refractive index, this indicates that the error depends not only on the optical depth but also on the refractive index. There is a possibility that the seasonal variation of $V_0$ by the IML method may also be related to the seasonal variation of the refractive index.

 In the current Improved Langley method, the refractive index is fixed. We used (1.5, −0.001) for all wavelengths as initial value of refractive index when using the Skyrad package. This value may not be appropriate. It is necessary to develop the method to determine $V_0$ while changing the refractive index in the future.

newFig. 9

[Figure]

Fig. 9 (a) scatter plot of optical thickness at 500nm and $\Delta V_0$ for 500nm channel, (b) scatter plot of $\Delta V_0$ and $W_0$ for 500nm channel, (c) time series of $\Delta V_0$ for 500nm channel in the period from January 2014 to December 2015 are shown. $\Delta V_0$ is the difference between $V_0$ determined by the IML method and $V_0$ interpolated from $V_0$ determined by inter-comparison with the POM-02 (Calibration Reference).

---

## Author Comment (AC3) · 18 Jun 2018

Reply to comments

We would like to thank you for reading our manuscript and commenting on it.

The comments are copied and shown below in italic.

*Comment.*

*Anonymous Referee #2*

*The paper describes in-field and laboratory calibrations of POM-02 sky radiometers used by SKYNET aerosol Network. The first method includes Langley-plot "zero airmass" intercept determination at high altitude Mauna Loa observatory for the POM-02 (Calibration reference) instrument with following calibration transfer to other instruments on-site (e.g., at Tsukuba site). The second on-site calibration method includes Improved Langley calibration (IML), without using reference instrument. Temperature effect on the calibration constant is shown to be important in the UV (340nm and 380nm) and shortwave infrared (2200nm) spectral channels. The temperature sensitivity varies for different instruments. The temperature effect on visible and NIR channels is generally small (< 0.5% for a typical temperature range).*

*The paper is of general interest for ground-based aerosol measurement community and could be published after major revision.*

*General comments:*

*The main manuscript should be clarified focusing on main conclusions, while supporting material (technical details, tables and plots) could be moved to the supplement.*

*Clarify calibration adjustment to account for changing sun-Earth distance.*

*English should be improved.*

*References need to be updated.*

*Figure quality needs improvements*

*==>*

Reply

  We wrote a lot of things, so the contents are discursive. However, one of the reviewers requested more explanations. In the first revision, we would like to keep it as it is now. Also, since there was nothing we wrote about our Langley method, we would like to write it in the main text.

  Before submitting the final version, we will receive corrections in English by native speaker of English.

We attach slightly enlarged figures to the revised manuscript.

*Specific comments:*

*Describe how spectral response functions were measured.*

==>

Reply

We have not measured the spectral response function of the filter by ourselves.

When we purchased POM-02, we requested the manufacturer to attach a response function as material.

The nominal specifications of the response functions are shown in Table 1.

Table ### Nominal filter specification

| Channel No. | Wavelength (nm) | FWHM(nm) | Max. Transmittance | Blocking | Blocking wavelength | Detector |
|---|---|---|---|---|---|---|
| – | 315(±0.6nm)* | 3.0(±0.6nm) | >30% | $1.0 \times 10^{-5}$ | 200 – 1200 nm | Si photodiode |
| 1 | 340(±0.6nm) | 3.0(±0.6nm) | >30% | $1.0 \times 10^{-5}$ | 200 – 1200 nm | Si photodiode |
| 2 | 380(±0.6nm) | 3.0(±0.6nm) | >30% | $1.0 \times 10^{-5}$ | 200 – 1200 nm | Si photodiode |
| 3 | 400(±0.6nm) | 10.0(±2.0nm) | >30% | $1.0 \times 10^{-5}$ | 200 – 1200 nm | Si photodiode |
| 4 | 500(±2.0nm) | 10.0(±2.0nm) | >30% | $1.0 \times 10^{-5}$ | 200 – 1200 nm | Si photodiode |
| 5 | 675(±2.0nm) | 10.0(±2.0nm) | >30% | $1.0 \times 10^{-5}$ | 200 – 1200 nm | Si photodiode |
| 6 | 870(±2.0nm) | 10.0(±2.0nm) | >30% | $1.0 \times 10^{-5}$ | 200 – 1200 nm | Si photodiode |
| 7 | 940(±2.0nm) | 10.0(±2.0nm) | >30% | $1.0 \times 10^{-5}$ | 200 – 1200 nm | Si photodiode |
| 8 | 1020(±2.0nm) | 10.0(±2.0nm) | >30% | $1.0 \times 10^{-5}$ | 200 – 3000 nm | Si photodiode |
| 9 | 1225(±2.0nm)** | 20.0(±2.0nm) | >30% | $1.0 \times 10^{-5}$ | 600 – 3000 nm | InGaAs photodiode |
| 10 | 1627(±2.0nm) | 20.0(±2.0nm) | >30% | $1.0 \times 10^{-5}$ | 600 – 3000 nm | InGaAs photodiode |
| 11 | 2200(±2.0nm) | 20.0(±2.0nm) | >30% | $1.0 \times 10^{-5}$ | 600 – 3000 nm | InGaAs photodiode |

FWHM : Full Width at Half Maximum

* : 315 nm channel is not used by JMA/MRI.

** : 1225 nm channel is used   by JMA/ MRI.

*Replace "near-infrared" with "shortwave infrared": for > 1-micron channels*

==>

Reply

We replaced "near-infrared" with "shortwave-infrared".

*Suggest replacing "SVA" with commonly used Field of View (FOV)*

==>

Reply

We use the term SVA for the magnitude value of FOV.

The term SVA was used in Nakajima et al. (1996), and it is familiar to users of POM-02.

We gave the following explanation to the term SVA after line 71.

"According to Nakajima et al. (1996), this paper uses SVA as a term representing the magnitude value of the field of view."

*Improved Langley method (IML) should be clearly explained – see comments L359-379*

==>

Reply

Please see below.

*L21: indicate temperature climatology in Tsukuba*

==>

Reply

We added location information and range of monthly mean temperature.

" at Tsukuba ( (36.05°N, 140.13°E), the range of monthly mean temperature 2.7 to 25.5° C)."

*L23: Is this accuracy at Tsukuba or Mauna Loa?*

==>

Reply

It is at Mauna Loa.

We rewrote the sentence.

*L25: quantify V0 uncertainty in UV-VIS-NIR and degradation (V0 time drift?)*

==>

Reply

We added the value of time degradation

"The degradation of $V_0$ for shorter wavelengths (−10 to −4% per year) was larger than that for longer wavelengths (−1 to nearly 0% per year)."

*L26: Clarify that this is accuracy of calibration transfer only during best stable atmospheric conditions. Indicate time intervals for calibration transfer*

==>

Reply

We rewrote as follows.

"The coefficient of variation (CV, standard deviation/mean) of $V_0$ transferred from the reference POM-02 was 0.1 to 0.5%. Here, the data was simultaneously taken every 1 minute on a fine day, and data with an airmass less than 2.5 were compared."

*L33: change to short infrared*

==>

Reply

We replaced "near-infrared" with "shortwave-infrared".

*L35: this sentence does not belong to the abstract*

==>

Reply

We rewrote sentences in line 31 to 37 as follows.

"The modified Langley method was attempted to calibrate the 940 nm channel using onsite measurement data. The difference from $V_0$ based on the Langley method of $V_0$ was better than 1% on selected stable and fine days. The General method was also attempted to calibrate the shortwave-infrared channels (1225, 1627, and 2200 nm) using onsite measurement data. The differences from $V_0$ based on the Langley method of $V_0$ were 0.8, 0.4 and 0.1% in December 2015, respectively."

*L37: Quantify accuracy for each channel.*

==>

Reply

Please see above.

*L59: Column average effective aerosol characteristics・・・*

==>

Reply

We replaced with "aerosol characteristics" with "column average effective aerosol characteristics".

*L68: add references*

==>

Reply

We add reference.

*L71: SVA is usually called Field of View (FOV)*

==>

Reply

We use the term SVA for the magnitude value of FOV.

The term SVA was used in Nakajima et al. (1996), and it is familiar to users of POM-02.

We gave the following explanation to the term SVA after line 71.

"According to Nakajima et al. (1996), this paper uses SVA as a term representing the magnitude value of the field of view."

*L74: Provide instrumental reference.*

==>

Reply

We add instrumental reference.

*L135: which temperature sensor is used to start the heater?*

==>

Reply

We added "near the rotating filter wheel" after "the temperature".

*L136: ": : : the instrument is heated: : :" – to what temperature? When does the heater stop ?*

==>

Reply

We added the following sentence after line 136.

"When the temperature exceeds the threshold, heating is stopped."

*L139: use "shortwave infrared"*

==>

Reply

We replaced "near-infrared" with "shortwave-infrared".

*L142: inside temperature[s]?*

==>

Reply

We replaced "inside temperature of instrument" with "temperature near the rotating filter".

*L146, Fig 1: Explain why if the temperature control setting was 20C , the inside temperature was 30C when the ambient temperature was 20C?*

==>

Reply

We added the following sentences to the explanation of Fig. 1 after line 145.

"Since heat is generated from the electric circuit inside the POM-02, the internal temperature exceeds 20 ℃ even if the ambient temperature is less than 20 ℃. The heater stops when the inside temperature of the POM-02 exceeds 20 ℃. However, since there is no cooling function, the temperature inside the POM-02 rises as the ambient temperature increases. When the ambient temperature is very low, the temperature does not rise to 20 ℃ because the heater capacity is insufficient. For example, when the ambient temperature was about −20 ℃, the internal temperature was about 0 ℃."

*L153: ∶∶∶ wavelengths shorter than 1020nm ∶∶∶*

==>

Reply

We rewrote it.

*L 157 "near-infrared region" – common name is "shortwave -infrared region"*

==>

Reply

We replaced "visible" with "visible and near-infrared"

And, we replaced "near-infrared" with "shortwave-infrared"

*L161: "change in the temperature less than 1.5%" -> "change in the instrument response less than 1.5%"?*

==>

Reply

We replaced "the temperature" with "the instrument response".

*L188, Fig.4: Specify units in Y axis , e.g. counts per second? – this is usually a large number: explain scaling.*

==>

Reply

The unit of data recorded in the file is A.

We redrew Figure 4.

*L197-248, Table 1: Explain units for calibration constant (V0) ? Table 1: Explain if correction for changing Sun-Earth distance was applied to daily V0s?.*

==>

Reply

We added unit.

We explained in the text that the calibration constant is the output of the radiometer to the extra-terrestrial solar irradiance at the mean earth-sun distance (1AU).

*L204 The [standard] error*

==>

Reply

SD/Mean is called coefficient of variation or relative standard deviation.

"ERR" is not appropriate.

We replaced "error" with "CV".

*L210 "is large :::" – quantify*

==>

Reply

We rewrote line 210 and 211 as follows.

"Based on the ratio of (GABS,NTPC)/(GBAS,TPC) (= (Case 3)/(Case 4)), the effect of the temperature dependence on the 340 and 2200 nm channels were about 3 and 5%, respectively."

*L222" without consideration of the temperature :::" –for MLO conditions only*

==>

Reply

We inserted the following sentence after line 223.

"The results shown here are the results obtained using the data taken at MLO."

*L233"was replaced" -> were replaced*

==>

Reply

We do not think that "The lens" is a plural form.

*L240: What are reasons for such large V0 changes ?*

==>

Reply

We are only users of POM-02. Not all information is received from the manufacturer of POM-02. Therefore, I do not know the reason for clear. We showed the fact that V0 tended

to change over time. Users of POM-02 should be aware of this.

*L244: It would be useful to show monthly V0 values (corrected for sun-Earth distance) in fig.5.*
==>
Reply
V0 shown in Fig. 5 is the output of the radiometer to the extra-terrestrial solar irradiance at the mean earth-sun distance (1 AU).

*L250: "Accuracy of [V0 calibration] transfer by direct solar measurement"*
==>
Reply
We rewrote it.

*L267 5. Improved Langley method - Add paragraph describing IML here*
==>
Reply
At the beginning of section 5, we moved most of section 5.2.

*L289 IML value[s]*
==>
Reply
We rewrote it.

*L295: delete "by"*
==>
Reply
We deleted "by".

*L330 layer -> atmospheric column optical thickness*
==>

Reply

We replaced "layer" with "atmospheric column".

*L322 5.2 Review of Improved Langley method – move this section up after L267*

==>

Reply

We moved most of this section to the beginning of section 5.

The description related to Figure 9 was moved to Section 5.1.

*L325 The solar direct irradiance at the surface [normal to the solar beam]*

==>

Reply

We inserted "normal to the solar beam".

*L330 zenith angle, and [tau] is the layer optical thickness. – total atmospheric optical thickness (Rayleigh plus aerosol plus gases)*

==>

Reply

We replace "layer optical thickness" with "total atmospheric optical depth"

*331 The single scattering by aerosol in the almucantar - replace "by aerosol plus molecular (Rayleigh) scattering in the almucantar "*

==>

Reply

We replaced "aerosol" with "aerosol and molecular ".

*L340 direct solar [voltage] measurement*

==>

Reply

We rewrote the sentence as follows.

"The sensor output for the direct solar measurement can be written as follows."

*L341: Equation (3) neglects forward scattered radiation into the FOV*

==>

Reply

We are writing in the text (line 342 and 343) that the contribution of scattered light in the field of view is ignored.

*L341-345: Equations (3) and (4) should include Earth-sun distance (see eq. (19) )*

==>

Reply

We included earth-sun distance in eq. (3) and (4).

We also rewrote eq. (1) and (2) and F0 is the value at 1 AU.

*L353: Explain how tau or tau_scat can be obtained independently from the V0 ?*

==>

Reply

tau and tau_sca can be obtained from eq. (5) and (6).

In the SKYRAD package, in the retrieval process for the Improved Langley method the single scattering and multiple scattering components are estimated by solving the radiative transfer equation. Once the single scattering component is retrieved, tau and tau_sca can be estimated (see eq. 10).

*L359:Explain how tau is obtained ?*

*L360: Explain how tau_scat is obtained?*

==>

Reply

The method for obtaining tau and tau_sca is the IML method.

For details, please see the original paper Tanaka et al. (1886) and Campanelli et al. (2004).

*L367: SVA ->FOV (common name)*

==>

Reply

See above.

*L368: ".. is the [radiometer output (voltage) due to ] direct solar irradiance [at the surface]*

==>

Reply

We inserted "radiometer output due to".

*L375: "Once the single scattering component is retrieved, m\*tau and m\*tau_scat are estimated" - Solving radiation transfer equation is only possible if tau, Phase function and tau_scat are known. Explain how tau, P(scat) and tau_scat are obtained?*

*What are introduced uncertainties due to assumptions about unknown aerosol refractive index, size distribution, modeling of aureole forward scattered radiation?*

==>

Reply

In this paper, we briefly explained the principle of IML method.

For details, please see Tanaka et al. (1986), Campanelli et al. (2004) and Skyrad package itself.

We rewrote sentences between line 373 and 378 as follows.

"In the SKYRAD package, given initial value of column particle volume size distribution $dV/d\log r$ and complex refractive indexes, $\tau$, $P(\cos\Theta)$, and $\omega_0$ are calculated. On the basis of these single scattering properties, the multiple scattering term (second term on the right side) in eq. (10) is evaluated, and the single scattering term (first term on the right sides) in eq. (10) can be obtained. The new $dV/d\log r$ is retrieved from the single scattering term in eq. (10) by the inversion scheme. Using the retrieved $dV/d\log r$, $\tau$, $P(\cos\Theta)$, and $\omega_0$ are calculated, and reconstruct the observed values, and then the error is calculated. Until the error satisfies convergence condition, the above procedure is iterated. In the above procedure, the complex refractive indexes for each channel are fixed and the measurement data with a scattering angle of less than 30 degrees are used."

*L379 "Once m*tau is obtained," – explain how tau is obtained before knowing calibration constant V0? Is another co-located radiometer used to derive tau?*
==>

Reply
Please see above.

*L380-381 do not use capital for single scattering albedo: W0*
==>

Reply

The single scattering albedo must be a value between zero and one. However, $W_0$ is frequently greater than 1. Therefore, it is only a constant for fitting. To distinguish between $\omega_0$ and $W_0$, $W_0$ was used.

We replace sentences between line 380 and 384 with the following sentences.

"$\ln V_0$ is determined by fitting to $\ln V = \ln V_0 - m\tau_{sca}/W_0$. Comparing this equation with

eq. (8), $W_0$ must be single scattering albedo. The single scattering albedo is defined as the ratio of the scattering coefficient to the extinction coefficient. Therefore, the single scattering albedo must be a value between zero and one. However, $W_0$ is frequently greater than 1. Therefore, it is only a constant for fitting. To distinguish between $\omega_0$

and $W_0$, $W_0$ was used."

*L384 "W0 is frequently greater than 1." – are these unphysical retrievals used for calibration?*
==>

Reply
There seems to be something wrong with the Skyrad Package procedure.
We focus only on pointing out that there are problems.

[Figure]

Solving the problem is a future task.

*L387 Figs. -> Fig.9*

==>

Reply

We fixed it.

*L389 "V0 values with errors less than 0.01" – Is this error in ln(V0) ?*

==>

Reply

We add the explanation of "error".

*L393, Fig9(c): In this plot was V0 corrected for the changing sun-Earth distance?*

==>

Reply

We redrew Fig. 9(c).

V0 is the value at the Earth-sun distance 1AU.

*L398 "⋮⋮are systematically overestimated". – please, clarify this statement*

==>

Reply

We delete sentence from line 397 to line 399.

*Ă˘nĂ˘n L414 " [and spectral response function of the ] radiometer are necessary"*

==>

Reply

We added "and spectral response function of the".

*L438: Table 4: Provide units for Vsun and V0*

==>

Reply

We added unit for Vsun and V0.

*L444-455: Fig 10: Compare with more recent sources of high spectral resolution extra-terrestrial solar irradiance, e.g.*
*https://www.cfa.harvard.edu/atmosphere/publications/Chance-Kurucz-solar2010-JQSRT.pdf*
*==>*

Reply

We added Chance and Kurucz (2010) to Fig. 10.

The value is a mean value weighted by the response function of a triangle with FWHM of 10nm.

*L466: which takes [into] account ∶∶∶*
*==>*

Reply

We fixed it.

*L488-490: Use tau_aer in Eq (19) and 490*
*==>*

Reply

We replaced tau with tau_aer.

*L492. "..is interpolated from the optical thicknesses at 870 and 1020 nm" – explain interpolation method, e.g. linear, power law?*
*==>*

Reply

We added the following sentence after 492.

"When interpolating $\tau_{aer}$ at 940 nm, $\tau_{aer}$ was assumed to be proportional to $\lambda^{-\alpha}$, where $\lambda$ is wavelength."

*L496: explain how R is calculated?*

==>

Reply

We showed reference.

Nagasawa, K. 1981: Tentai no ichi keisan (Position calculation of celestial bodies), Chjin Shokan, p. 239 (in Japanese).

The literature is written in Japanese, but it is often used by Japanese researchers.

We inserted the following sentence in line 497.

   "For example, $R$ can be calculated with a simplified formula by Nagasawa (1981), $m$ can be calculated with the formula by Kasten and Young (1989), and $\tau_R$ can be calculated with the formula by Asano et al. (1983)."

*L497-499: explain how coefficients a and b were calculated?*

==>

Reply

See Apendix.

Details of the method for determining the coefficients $a$ and $b$ are described in Uchiyama et al. (2014).

*L504-505: explain units for calibration coefficients?*

==>

Reply

We added unit.

*L548: use tau_aer*

==>

Reply

We replaced tau with tau_aer.

*L640: "seasonal variation of 1 to 3%." – Correcting for sun-Earth distance ?*

==>

Reply

In this paper, calibration constant V0 is the output of the radiometer to the extra-terrestrial solar irradiance at the mean earth-sun distance (1 AU). And, when temperature correction is applied to the sensor output, it is the value at the reference temperature.

*Technical comments:*
*L559-560: Equations (24) and (25) can be combined.*
==>

Reply
We deleted eq. (24).

*L585: ": : : is an alternative to the Langley method." – extension of Langley method?*
==>

Reply
This expression was not appropriate.
We rewrote this sentence as follows.
"The method shown here is the next best solution."

*L629: " The annual variation of the calibration constants: : :" – The long-term changes*
==>

Reply
We replaced "annual variation" with "long-term changes".

*Fig 1 caption: "inside temperature[s]"*
==>

Reply
We replaced "temperature" with "temperatures".

*Fig.4. Check the Y units: be counts per second? What is the scaling factor?*
==>

Reply

The unit of sensor output is A (Ampere).

*Fig.5. Show monthly V0 values to check V0 seasonal dependence*
==>
Reply
The V0 values shown in Fig. 5 are values determined on the basis of measurements made at MLO once a year. There are no monthly values. We cannot show the monthly V0 values. The change in the V0 values is smooth and can be interpolated.

*Fig. 8 Too small axis labels. Suggest scaling Y axis for clarity*
==>
Reply
We enlarged the figures.

---

## Editor Decision (ED1)

The paper has been significantly improved. A second review round indicates minor, but numerous, editing corrections. In addition to those changes, the following three issues should be addressed.

1. The abstract contains too many technical details. It should be re-written including only high level conclusions. The technical details should be moved to conclusions section and supplement. 2

2. The units of the sensor output and calibration constant (V0) need to be defined and indicated on V0 plots and captions. Plotting ln(v0) on Y axis in Figs. 4-5 and 8-9 is suggested.

3. Additional justification should be given to support the use of the improved Langley Method. The IML involves complex iterative inversion scheme, implemented in SKYRAD retrievals package, which requires single and multiple scattering radiative transfer calculations to iteratively estimate particle column volume size distribution, and calculate single scattering albedo and phase function. These inversions require assumptions about particle sphericity, fixed complex refractive index, and surface reflectance. It is difficult to estimate the accuracy of the retrieved effective aerosol parameters. For example, retrieved single scattering albedo could exceed unity (Line 558 and Fig. 9b). The authors admit that the fixed value of column effective refractive index (1.5 – 0.001i) used in SKYRAD inversion "may not be appropriate" (L641). Moreover, these parameters are irrelevant if the goal is simply estimation of the calibration constant (V0), which requires only knowledge of the aerosol extinction optical depth and surface pressure, assuming gaseous absorption is negligible. Figure 8 clearly shows much higher noise of the IML method compared to calibration transfer from co-located reference instrument. The authors should justify using IML compared to the traditional calibration methods (e.g., transferring calibration from the reference POM-02, using calibrated light source), and estimate its uncertainties, e.g., in case of non-spherical dust particles.

A major concern is that the SKYRAD package combines calibration procedure with the optical inversion scheme, which involves many highly uncertain a-priori assumptions. It is always preferable to keep the calibration step (i.e., determining V0) independent from the inversion step.

Minor Technical suggestions:

L27,28. Indicate, which wavelengths?
L48-49: add references for health effects of aerosols
51-54 : re-word
77: Add V0 after "calibration constant, V0, …". Provide units for the calibration constant here, e.g. counts/sec, voltages, etc.

98: Add: precision of the calibration constant [transfer] obtained from …

122 Add units: is located at an elevation of 3397.0 [meters]

157 where V (T ) is the sensor output [voltage] - ?

159 Therefore, the measured V (T ) is corrected [using equation (1) ]
161: equation (2) is the same as equation (1) and could be deleted.

218 ".. is large? for this POM-02" – the temp sensitivity numbers are roughly the same or smaller than for the calibration reference POM-02 given in previous paragraph.

223 The temperature dependence of the detector sensitivity – suggest: detector response

224 specifications of the detector – which specifications? Provide reference.

241 comparing the side-by-side – remove "the"

254 When the extinction coefficient is divided? - is defined?

256 Introducing the normal? optical thickness (or optical depth). – replace with vertical optical thickness

262 is the airmass for the i-th - "th" should be subscript

290, Remove the first part of Eq (10), which is the same as Eq (9)

304 depth[s]

305 If the sensor output [voltage]

357 is proportional to the sum of the line absorption strength[s]

417 measurements for the calibration at MLO were being conducted. – remove "being"

436 Figure 5 shows the annual [multiyear] variation of the calibration constants - in what units?

450: annual variation -> interannual variations?

451 from 2009 to 2016 - There is no 2016 in the plots

484-485: "The results of the comparison showed that the JMA's POM-02 met the WMO criterion (WMO 2005)." Please state the accuracy criteria of the WMO here.

498 The single scatter[ed radiance] …
518 Equation 25 is the same as Equation (15)

520 "If m, mτ, and , mτscat can be obtained …" – If this is the case, ln(V0i) can be simply calculated using equations (25)-(26) for each individual measurement (Vi) and averaged for any time period.

526-529: The SKYRAD retrieved/assumed effective parameters, such as single scattering albedo (SSA) and scattering aerosol optical thickness are irrelevant if the goal is determination of the calibration constant (V0), which requires only knowledge of the aerosol extinction optical depth and surface pressure, assuming gaseous absorption is negligible.

540-552: Clarify if SKYRAD inversion procedure assumes spherical particles only?

550 "..procedure, the complex refractive indexes for each channel are fixed" – clarify what ref. index values are assumed? How they compare with the AERONET retrieved values?

554: "Comparing this equation with eq. (26)," - Why not using simpler equation (25) directly?

577: " In Fig. 8, the calibration constants .." – Explain V0 units. Change to as Y-axis ln(v0) to express % changes directly and comparable across all spectral channels.

Fig. 8 shows much higher noise of the IML method compared with calibration transfer method (red points).
This should be clearly stated in conclusions and abstract.

733. pwv is PWV – explain abbreviation
743. "V is the measurement value" – clarify the units, i.e., voltage , count rate, etc.
749 be fitted by a linear function of mb – Left hand side also includes m

755 "2.2973Å~10−4 A". – what is A?

761 (2.2973Å~10−4/2.3364Å~10−4 − 1 = −0.0167) – delete
762 (2.2954Å~10−4/2.3157Å~10−4 − 1 = −0.0087) – delete

Lines 780-782: "....the calibration constant of the 940 nm channel could be determined by applying the above-mentioned method on a suitable stable and fine day at the observation site." Please define how stable the water vapor needs to be over the interval of Langley observations for this technique to be accurate to within ~1%.

894. The changes in the 340 nm channel were −10% per year" – this is very large degradation rate and requires recommendations for upgrading this channel

1195-1198: Define units of V/V0
1208: Define V0 units

1214: 9(c): Define V0 units

Table 2: (unit is A) – Explain meaning of A (Ampere?)

---

## Author Response (AR2)

Reply to comments

We would like to thank you for reading our manuscript and commenting on it.

The comments are copied and shown below in italic.

*The paper has been significantly improved. A second review round indicates minor, but numerous, editing corrections. In addition to those changes, the following three issues should be addressed.*

*1. The abstract contains too many technical details. It should be re-written including only high level conclusions. The technical details should be moved to conclusions section and supplement.*

==>

The abstract of the first version included technical details. Since one of the reviewers commented on this point, in the revised version, technical details were removed according to the comment. Is further revision necessary?

*2. The units of the sensor output and calibration constant (V0) need to be defined and indicated on V0 plots and captions. Plotting ln(v0) on Y axis in Figs. 4-5 and 8-9 is suggested.*

==>

We added a description of the unit of V0 to the main text and the figure captions.

The Langley method is relative calibration, not absolute calibration. Therefore, we do not have to worry about units.

The sensor output depends on the instrument; voltage, current, digital count value etc.

Since the reviewers repeatedly points out the unit, we added an explanation and unit of sensor output of POM-02 in the revised version.

*3. Additional justification should be given to support the use of the improved Langley Method.*

*The IML involves complex iterative inversion scheme, implemented in SKYRAD retrievals package, which requires single and multiple scattering radiative transfer calculations to iteratively estimate particle column volume size distribution, and calculate single scattering albedo and phase function. These inversions require assumptions about particle sphericity, fixed complex refractive index, and surface reflectance. It is difficult to estimate the accuracy of the retrieved effective aerosol parameters. For example, retrieved single scattering albedo could exceed unity (Line 558 and Fig. 9b). The authors admit that the fixed value of column effective refractive index*

*(1.5 – 0.001i) used in SKYRAD inversion "may not be appropriate" (L641). Moreover, these parameters are irrelevant if the goal is simply estimation of the calibration constant (V0), which requires only knowledge of the aerosol extinction optical depth and surface pressure, assuming gaseous absorption is negligible. Figure 8 clearly shows much higher noise of the IML method compared to calibration transfer from co-located reference instrument. The authors should justify using IML compared to the traditional calibration methods (e.g., transferring calibration from the reference POM-02, using calibrated light source), and estimate its uncertainties, e.g., in case of non-spherical dust particles.*

==>

We do not necessarily recommend using the IML method.

We conduct observations by POM-02 independently of SKYNET. In JMA / MRI, the calibration reference POM-02 is calibrated by normal Langley method using the data taken at MLO, and the calibration constants of the other POM-02s for the continuous measurements are transferred from the reference POM-02. We have not used the IML method.

The objectives in this study are to investigate the current status of and problems with the sky radiometer POM-01 and POM-02.

We showed that the calibration constant determined by IML method has a seasonal variation of 1 to 3%, and in some cases, the maximum difference reaches about 5%. This indicates that the 2% error in the calibration constant is not significant in a turbid atmosphere, but it is significant in a clear atmosphere, such as in polar and ocean regions. Furthermore, there is a possibility that the seasonal variation of the calibration constant causes an artificial seasonal variation in the retrieved parameters.

However, many POM-01 and POM-02 users have been measuring without the periodic calibration by normal Langley method, and a lot of data has been accumulated. To use these data, we have to rely on the calibration with IML method.

*A major concern is that the SKYRAD package combines calibration procedure with the optical inversion scheme, which involves many highly uncertain a-priori assumptions. It is always preferable to keep the calibration step (i.e., determining V0) independent from the inversion step.*

==>

We also have the same idea, and we consider the calibration constant V0 to be determined independently from the inversion scheme.

As said above, however, many POM-01 and POM-02 users have been measuring without

the periodic calibration by normal Langley method, and a lot of data has been accumulated. To use these data, we have to rely on the calibration with IML method.

*Minor Technical suggestions:*

*L27,28. Indicate, which wavelengths?*
==>
We inserted wavelength.

*L48-49: add references for health effects of aerosols*
==>
We added references.

*51-54 : re-word*
==>
Is this sentence long?
We rewrote this sentence as follows.
"Therefore, measurement networks covering an extensive area on the ground and from space have been developed and established to determine the spatiotemporal distribution of aerosols."

*77: Add V0 after "calibration constant, V0, …". Provide units for the calibration constant here, e.g. counts/sec, voltages, etc.*
==>
We inserted V0.
The unit of V0 depends on the instrument. The sensor output is voltage, current, digital count value, etc. Here, we do not specify the instrument. So, we do not provide the unit.

*98: Add: precision of the calibration constant [transfer] obtained from …*
==>
We inserted "transfer".

*122 Add units: is located at an elevation of 3397.0 [meters]*
==>
We inserted "meters".

*157 where V (T ) is the sensor output [voltage] - ?*

==>

We inserted the following sentence;

 "In the case of POM-02, the sensor output is current, and the unit is Ampere (A)."

*159 Therefore, the measured V (T ) is corrected [using equation (1) ]*

*161: equation (2) is the same as equation (1) and could be deleted.*

==>

We rewrote the sentence, and deleted eq. (2).

*218 ".. is large? for this POM-02" – the temp sensitivity numbers are roughly the same*
*or smaller than for the calibration reference POM-02 given in previous paragraph.*

==>

We made a mistake.

We rewrote the sentence as follows..

"The temperature dependence of the sensor output in the 380 nm (2200 nm) channel for this POM-02 is larger (smaller) than that for the calibration reference POM-02."

*223 The temperature dependence of the detector sensitivity – suggest: detector response*

==>

Yes.

*224 specifications of the detector – which specifications? Provide reference.*

==>

"specifications" means "specifications data sheet".

We replaced "specifications" with "specifications data sheet".

And the address of the web is shown as a reference.

https://www.hamamatsu.com/resources/pdf/ssd/s1336_series_kspd1022e.pdf

https://www.hamamatsu.com/resources/pdf/ssd/g12183_series_kird1119e.pdf

*241 comparing the side-by-side – remove "the"*

==>

We removed "the".

*254 When the extinction coefficient is divided? - is defined?*

==>

We rewrote the sentence as follows,

"When the extinction coefficient is composed of several components,"

*256 Introducing the normal? optical thickness (or optical depth). – replace with vertical optical thickness*

==>

We replaced "normal" with "vertical".

In the textbook written by Liou, "normal" is used.

*262 is the airmass for the i-th - "th" should be subscript*

==>

We replaced "i-th" with "i-th".

*290, Remove the first part of Eq (10), which is the same as Eq (9)*

==>

We removed the first part of Eq (10).

*304 depth[s]*

==>

We replaced "depth" with "depths".

*305 If the sensor output [voltage]*

==>

The sensor output depends on the instrument: voltage, current, digital count value, etc.

Here, we do not specify the instrument. So, we do not provide the unit.

*357 is proportional to the sum of the line absorption strength[s]*

==>

We replaced "strength" with "strengths".

*417 measurements for the calibration at MLO were being conducted. – remove "being"*

==>

We removed "being".

*436 Figure 5 shows the annual [multiyear] variation of the calibration constants - in*

*what units?*

==>

We inserted "multiyear".

The unit is shown in the Fig. 5 caption.

*450: annual variation -> interannual variations?*

==>

We replaced "annual" with "interannual".

*451 from 2009 to 2016 - There is no 2016 in the plots*

==>

When we revised the manuscript and figure, we added 2016 data.

*484-485: "The results of the comparison showed that the JMA's POM-02 met the WMO criterion (WMO 2005)." Please state the accuracy criteria of the WMO here.*

==>

We inserted the following sentence.

"The WMO criterion for the absolute differences of all instruments compared to the reference is defined as follows: "95% of the measured data has to be within 0.005±0.001/m" (where m is the airmass)."

*498 The single scatter[ed radiance] …*

==>

We replace "scattering" with "scattered radiance".

*518 Equation 25 is the same as Equation (15)*

==>

The eqs. (25) and (26) help to understand the contents of the next paragraph.

We combined equations (25) and (26) together and left both equations.

*520 "If m, mτ, and , mτscat can be obtained …" – If this is the case, ln(V0i) can be simply calculated using equations (25)-(26) for each individual measurement (Vi) and averaged for any time period.*

==>

Here, we just explain how V0 is determined in the SKYRAD package.

*526-529: The SKYRAD retrieved/assumed effective parameters, such as single scattering albedo (SSA) and scattering aerosol optical thickness are irrelevant if the goal is determination of the calibration constant (V0), which requires only knowledge of the aerosol extinction optical depth and surface pressure, assuming gaseous absorption is negligible.*

==>

As said above, here we just explain how V0 is determined in the SKYRAD package.

In this paper, the accuracy was evaluated by comparing V0 determined by the IML method with V0 determined by the normal Langley method. We think that there are some problems with the IML method.

540-552: Clarify if SKYRAD inversion procedure assumes spherical particles only?

==>

We inserted the following sentence after "….are calculated"

"assuming the spherical homogeneous particle".

We developed another retrieval software based on the spheroid model. But, this software is not officially adopted by the SKYNET group.

Kobayashi E, A. Uchiyama, A. Yamazaki, R. Kudo, 2010: Retrieval of Aerosol Optical Properties Based on the Spheroids Model. J. Meteor. Soc. Japan, 88, 847-856, Doi:10.2151/jmsj.2010-505.

*550 "..procedure, the complex refractive indexes for each channel are fixed" – clarify what ref. index values are assumed? How they compare with the AERONET retrieved values?*

==>

The values of the complex refractive indexes for each channel can be given by the SKYRAD package user. The values that we used are described in section 5.2.

The objective of this paper is not the comparison with AERONET.

Che et al (2008) compared aerosol optical properties by a PREDE skyradiometer and CIMEL sunphotometer over Beijing, China.

H. Che, G. Shi, A. Uchiyama, A. Yamazaki, H. Chen, P. Goloub, and X. Zhang, 2008: Intercomparison between aerosol optical properties by a PREDE skyradiometer and CIMEL sunphotometer over Beijing, China, *Atmos. Chem. Phys.*, **8**, 3199–3214.

*554: "Comparing this equation with eq. (26)," - Why not using simpler equation (25) directly?*

==>

As said above, here we are just explaining how V0 is determined in the SKYRAD package. We are not in a position to answer this question. Because, we did not develop the SKYRAD package.

We also want to know why eq. (26) is used, but there is no written material.

In this paper, we only evaluated the accuracy of V0 obtained by the method currently used.

*577: " In Fig. 8, the calibration constants .." – Explain V0 units. Change to as Y-axis ln(v0) to express % changes directly and comparable across all spectral channels. Fig. 8 shows much higher noise of the IML method compared with calibration transfer method (red points). This should be clearly stated in conclusions and abstract.*
==>

We added an explanation of the unit of V0 in the figure caption.

I appreciate your advice. However, we want to keep the figure as it is.

We added the following sentences to the main text and abstract.

"Furthermore, Fig. 8 shows much higher noise of the IML method compared with calibration transfer method"

"Furthermore, the calibration constants determined by the IML method had much higher noise than those transferred from the reference."

*733. pwv is PWV – explain abbreviation*
==>

"PWV" has already been explained above.

*743. "V is the measurement value" – clarify the units, i.e., voltage , count rate, etc.*
==>

The unit of sensor output depends on the instrument. The sensor output is voltage, current, digital count value, etc. Here, we do not specify the instrument. So, we did not provide the unit. However, we applied this method just below. Therefore, we added the following sentence.

 "In the case of POM-02, the sensor output is current, and the unit of the measurement value V is Ampere (A)."

*749 be fitted by a linear function of mb – Left hand side also includes m*
==>

The airmass "m" is not an unknown. We can calculate the value of "m" from the solar

zenith angle.

*755 "2.2973Å~10−4 A". – what is A?*
*==>*

"A" is Ampere.
One of the reviewers requested to enter the unit repeatedly, so I entered the unit.
The Langley method is relative calibration, not absolute calibration. Therefore, we do not have to worry about units.
The unit of the current is ampere and the notation in the SI unit system is "A".

*761 (2.2973Å~10−4/2.3364Å~10−4 − 1 = −0.0167) – delete*
*==>*削除した。
We delete it.

*762 (2.2954Å~10−4/2.3157Å~10−4 − 1 = −0.0087) – delete*
*==>*削除した。
We delete it.

*Lines 780-782: "….the calibration constant of the 940 nm channel could be determined by applying the above-mentioned method on a suitable stable and fine day at the observation site." Please define how stable the water vapor needs to be over the interval of Langley observations for this technique to be accurate to within ~1%.*
*==>*

We applied Langley method to data in the airmass range between 2 and 6. The range of airmass is 2 to 5 in AERONET (Holben et al 1998) and 2 to 7 in Schmid and Wehrli (1995). Therefore, a stable interval of 1 to 2 hours is necessary. The stable condition of the atmosphere depends on the observation site. In Tuskuba, the outbreak of clod airmass from the continent may continue for a few days in winter. In such a case, the atmospheric change is small. In this study, this method was applied on such a day. As described below this sentence, it is an important point that the quality of the Langley plot can be checked by an analysis of the residuals.
We added the following sentences.
"We applied Langley method to data in the airmass range between 2 and 6. Therefore, a stable interval of 1 to 2 hours is necessary."

*894. The changes in the 340 nm channel were −10% per year" – this is very large*

*degradation rate and requires recommendations for upgrading this channel*

==>

As can be seen from Fig. 5, the degradation of V0 became smaller after replacing the lens in 2013. The manufacturer may have upgraded the lens.

We added the following sentence in section 4.2.

"After replacing the lens in 2013, the degradation of the 340 and 380 nm channels became smaller. The manufacturer of the skyradiometer may have upgraded the lens."

*1195-1198: Define units of V/V0*

==>

The units of V and V0 are the same. Therefore, V/V0 has no unit.

*1208: Define V0 units*

==>

We added a description of the unit of V0.

*1214: 9(c): Define V0 units*

==>

We added a description of the unit of V0.

*Table 2: (unit is A) – Explain meaning of A (Ampere)*

==>

We added a description of the unit of V0.